# *Thermosynechococcus* switches the direction of phototaxis by a c-di-GMP-dependent process with high spatial resolution

Daisuke Nakane[1]*[†], Gen Enomoto[2]*[†], Heike Bähre[3], Yuu Hirose[4], Annegret Wilde[2,5]*, Takayuki Nishizaka[6]

[1]Department of Engineering Science, Graduate School of Informatics and Engineering, The University of Electro-Communications, Tokyo, Japan; [2]Institute of Biology III, University of Freiburg, Freiburg, Germany; [3]Research Core Unit Metabolomics, Hannover Medical School, Hannover, Germany; [4]Department of Applied Chemistry and Life Science, Toyohashi University of Technology, Toyohashi, Japan; [5]BIOSS Centre for Biological Signaling Studies, University of Freiburg, Freiburg, Germany; [6]Department of Physics, Gakushuin University, Tokyo, Japan

**\*For correspondence:**
dice-k@uec.ac.jp (DN);
gen.enomoto@biologie.uni-freiburg.de (GE);
annegret.wilde@biologie.uni-freiburg.de (AW)

[†]These authors contributed equally to this work

**Competing interest:** The authors declare that no competing interests exist.

**Abstract** Many cyanobacteria, which use light as an energy source via photosynthesis, show directional movement towards or away from a light source. However, the molecular and cell biological mechanisms for switching the direction of movement remain unclear. Here, we visualized type IV pilus-dependent cell movement in the rod-shaped thermophilic cyanobacterium *Thermosynechococcus vulcanus* using optical microscopy at physiological temperature and light conditions. Positive and negative phototaxis were controlled on a short time scale of 1 min. The cells smoothly moved over solid surfaces towards green light, but the direction was switched to backward movement when we applied additional blue light illumination. The switching was mediated by three photoreceptors, SesA, SesB, and SesC, which have cyanobacteriochrome photosensory domains and synthesis/degradation activity of the bacterial second messenger cyclic dimeric GMP (c-di-GMP). Our results suggest that the decision-making process for directional switching in phototaxis involves light-dependent changes in the cellular concentration of c-di-GMP. Direct visualization of type IV pilus filaments revealed that rod-shaped cells can move perpendicular to the light vector, indicating that the polarity can be controlled not only by pole-to-pole regulation but also within-a-pole regulation. This study provides insights into previously undescribed rapid bacterial polarity regulation via second messenger signalling with high spatial resolution.

## Editor's evaluation

In this fascinating study, the authors explore the cellular and molecular bases of phototaxis in Cynaobacteria. It is shown that phototaxis is a highly efficient Spatio-temporal response involving "within a pole regulation" of retractile Type-IV pili. This unique regulation involves the interplay of three possible photoreceptors that regulate the intracellular concentration of c-di-GMP in response to green to blue light transitions.

**eLife digest** Cyanobacteria, like plants, grow by capturing energy from sunlight. But they have an advantage over their leafy counterparts: they can explore their environment to find the type of light that best suits their needs. These movements rely on hook-like structures, called type IV pili, which allow the cells to pull themselves forward. The pili are usually located at the opposite poles of a rod-shaped cell, allowing the bacteria to move along their longer axis. Yet, the molecular mechanisms that allow cyanobacteria to react to the light are poorly understood.

To explore these processes in more detail, Nakane, Enomoto et al. started by shining coloured lights on the rod-shaped cyanobacteria *Thermosynechococcus vulcanus*. This revealed that the cells moved towards green light but reversed rapidly when blue light was added. The behaviour was disrupted when the genes for three light-sensing proteins were artificially switched off. These molecular players act by changing the levels of cyclic di-GMP, a signalling molecule that may interact with type IV pili.

The experiments also showed that *T. vulcanus* cells were not only moving along their longer axis, but also at a right-angle. This observation contrasts with how other rod-shaped bacteria can explore their environment. A closer look revealed that the cyanobacteria could perform these movements by making asymmetrical adjustment to the way that pili at each pole were working. Further research is now needed to more finely dissect the molecular mechanisms which control this remarkable type of motion.

## Introduction

Cyanobacteria are phototrophic microorganisms, and optimal light conditions are crucial for efficient photosynthesis. Therefore, several cyanobacterial strains are able to move either into their preferred light habitat or away from a harmful or stressful environment (*Wilde and Mullineaux, 2017*). This decision-making process is based on sensing the light direction and light intensity and quality on a short time scale. Moderate red light is used as the preferred energy source for oxygenic photosynthesis, while strong light or UV light causes cell damage (*Latifi et al., 2009*). However, how cyanobacterial cells rapidly switch the direction of movement remains unclear.

Cyanobacterial movement is usually driven by type IV pili (T4P) (*Wilde and Mullineaux, 2017*), a general bacterial molecular machine. This machine enables cellular movement by repeated cycles of extension and retraction of the pili (*Chang et al., 2016*; *Craig et al., 2019*). T4P are often localized at the cell poles in rod-shaped bacteria, and their localization at a certain pole is dynamically controlled to achieve directional movement (*Talà et al., 2019*). Bacteria, such as *Pseudomonas*, exhibit chemotactic behaviour to activate T4P at the leading pole on a time scale of hours. Consequently, the longer axis of cells is roughly aligned in parallel along the chemical gradient (*Oliveira et al., 2016*). Cyanobacterial cells make a decision of directional movement in a few minutes upon light sensing by dedicated photoreceptors (*Nakane and Nishizaka, 2017*). Once the lateral light stimulus is applied to the cells, they detect the orientation of a light source using their own cell body as an optical lens. In the coccoid *Synechocystis* sp. PCC 6803 (*Synechocystis*), the light is focused at the cell envelope, thereby generating a light spot opposite to the light source. Such micro-optic effects have been described in cyanobacteria of different shapes (*Nakane and Nishizaka, 2017*; *Schuergers et al., 2016*). Yang et al. suggested that the rod-shaped cyanobacterium *Synechococcus elongatus* UTEX 3055 also utilizes the micro-optics effect to sense directional light by the polarly localized photoreceptor PixJ (*Yang et al., 2018*). However, T4P-dependent cell behaviour has not been clarified at the single-cell level in rod-shaped cyanobacteria (*Yang et al., 2018*). Furthermore, since most of the knowledge of phototaxis is derived from coccoid-shaped *Synechocystis*, how the polarity of the cell structure is involved in phototaxis regulation is not clear.

The nucleotide second messenger molecule cyclic dimeric GMP (c-di-GMP) is the critical molecule governing bacterial motility as the master regulator of bacterial lifestyle transitions (*Hengge, 2020*; *Jenal et al., 2017*). The high intracellular concentration of c-di-GMP is universally implicated in the repression of motility and induction of sessile multicellular community development. A decrease in c-di-GMP levels is often required for optimum motile behaviour involving T4P or flagella (*Jenal et al., 2017*). The diguanylate cyclase activity mediated by GGDEF domains is necessary to produce

c-di-GMP, whereas EAL and HD-GYP domains harbour phosphodiesterase activities to degrade c-di-GMP. These domains are often combined to signal sensory domains (*Agostoni et al., 2013*), enabling bacterial cells to integrate multiple environmental information into cellular c-di-GMP levels to orchestrate various cellular responses to accomplish complex lifestyle transitions.

In this study, we establish a microscopy setup to analyse the phototaxis of the rod-shaped thermophilic cyanobacterium *Thermosynechococcus vulcanus* using live cells at the single-cell level. We dissect the contributions of different light colours and evaluate the functions of each putative photoreceptor gene. We show that both positive and negative phototaxis in *T. vulcanus* were controlled by a specific green-to-blue light ratio. Furthermore, we provide evidence that the reversion of phototaxis is mediated by photoreceptors and their activity in the synthesis or degradation of c-di-GMP. The asymmetric distribution of T4P at both cell poles achieves directional movement with a random orientation of the cells along their optical axis. We suggest a within-a-pole regulation of polarity that governs the directional movement of *T. vulcanus* and requires signalling at high spatiotemporal resolution. The light dependency of the phototactic response is consistent with a proposed model of *T. vulcanus* behaviour in a mat, which is their natural habitat.

## Results

### Control of positive and negative phototaxis

*T. vulcanus* used in this study was originally derived from the strain at the culture collection of NIES-2134 (http://mcc.nies.go.jp/). Since the original strain exhibited a heterogeneous phenotype of phototaxis under optimal growth conditions, a clone showing clear positive phototaxis in moderate light was isolated (*Enomoto et al., 2018*), and its complete genome was sequenced (AP018202). This clone was used as wild type (WT) in the following experiments. The strain also showed negative phototaxis when we applied a strong light stimulus. This bidirectional phototaxis was visualized as colony migration on a BG11 agar plate in a long-term observation for 8 hr (*Figure 1A*), consistent with previous data based on a closely related strain (*Kondou et al., 2001*).

To observe a single-cell trajectory during phototaxis on a short time scale, we constructed an optical setup that allowed us to stimulate the cell with lateral light on a microscope stage, which was heated at a temperature of 45°C to observe the physiological cell behaviour (*Figure 1B,C*). The position of the rod-shaped cell was visualized by near-infrared light through a bandpass filter with a centre wavelength of 850 nm and a full width at half maximum of 40 nm from a halogen lamp with a fluence rate of 1 μmol m$^{-2}$ s$^{-1}$, which was confirmed to have no effect on motility (*Figure 1D,E*; *Kondou et al., 2001*; *Nakane and Nishizaka, 2017*). When we applied a lateral white illumination of 20 μmol m$^{-2}$ s$^{-1}$, the cells started to exhibit directional movement towards the light source in a few minutes (*Figure 1C, top*). On the other hand, the cells showed directional movement away from the light source upon strong white light illumination of 500 μmol m$^{-2}$ s$^{-1}$ (*Figure 1C, bottom*). The rose plot, a round histogram that simultaneously presents the number of occurrences and direction, indicates that lateral light illumination induced both positive and negative phototaxis (*Figure 1D*). The displacement of cell movement along the optical axis on a glass surface was calculated to be 20–50 μm min$^{-1}$ (*Figure 1E*), which is 5–10 times faster than that of *Synechocystis* or *S. elongatus* strain UTEX 3055 (*Nakane and Nishizaka, 2017*; *Yang et al., 2018*). These data clearly show that the positive and negative phototaxis is controllable on this microscopic setup. Therefore, we started with a more detailed quantitative analysis of the cell movement of *T. vulcanus*.

### Wavelength dependence of phototaxis

We set up five light emitting diodes (LEDs) with various wavelengths as lateral light sources to stimulate the phototaxis of cells under the microscope (*Figure 2A*). We used LEDs for blue, teal, green, red, and far-red light with peak wavelengths at 450, 490, 530, 625, and 730 nm, respectively, and spectral bandwidths in a range of 15–40 nm (*Figure 2B*). When we applied monochromatic illumination with green or teal light at a rate of 70 μmol m$^{-2}$ s$^{-1}$, the cells showed directional movement within a few minutes (*Figure 2C* and *Video 1*). The cell displacement along the optical axis was 30 μm min$^{-1}$ in the green and 20 μm min$^{-1}$ in the teal (*Figure 2—figure supplement 1*). Monochromatic illumination with blue or red light induced the fast formation of small aggregates that moved randomly (*Figure 2C*–*Figure 2—figure supplement 1*, and *Video 2*). The previously described macroscopic floc formation

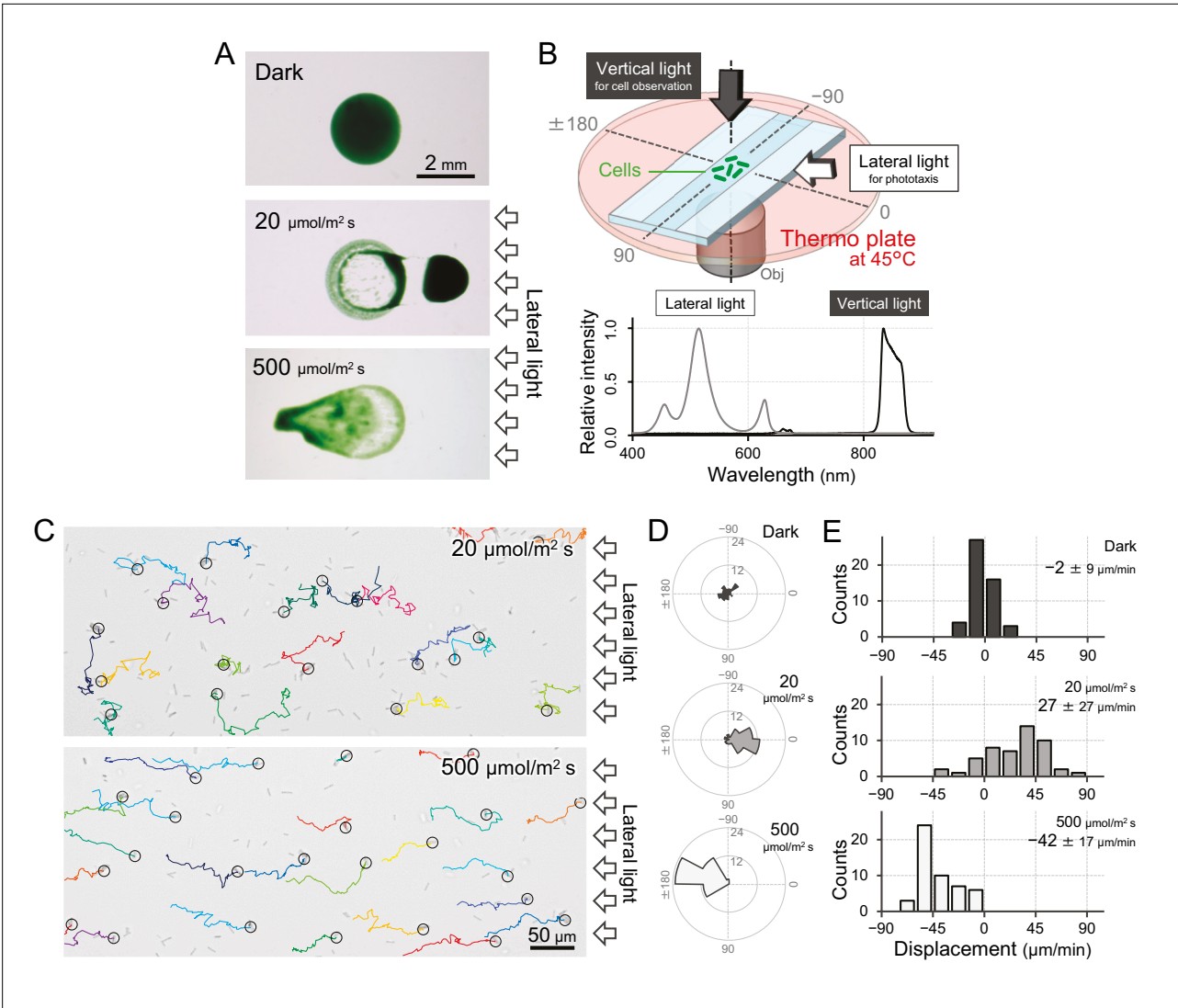

**Figure 1.** Positive and negative phototaxis of *T. vulcanus*. (**A**) Phototaxis on agar plates. Images were taken 8 hr after spotting the cell suspension. (**B**) Diagram of the experimental setup to visualize single cellular behaviour under optical microscopy. The glass chamber was heated at 45°C with a thermoplate on a microscope stage. Vertical and lateral light sources were used for cell observation and stimulation of phototaxis, respectively. Light spectra are presented at the bottom. (**C**) Bright-field cell image and their moving trajectories for 120 s (colour lines) on a glass surface. The cell at the start position of a trajectory is marked by the black circle. The white arrows on the right side of the image represent the direction of the light axis. (**D**) Rose plots under dark, weak light, and strong light illumination. The moving direction of a cell that translocated more than 6 µm min⁻¹ was counted. Angle 0 was the direction towards the lateral light source ($N = 50$ cells). (**E**) Histograms of the cell displacement along the lateral light axis. The cell displacement for a duration of 1 min was measured at 4 min after lateral illumination was turned on ($N = 50$ cells). Cell movements towards the light source are shown as a positive value.

The online version of this article includes the following source data for figure 1:

**Source data 1.** Rose plots.

**Source data 2.** Cell displacement.

of *T. vulcanus*, which involves the induction of cellulose synthesis (*Kawano et al., 2011*), is a long-term response that is referred to as cell aggregation. For clarity, in the current study, we call the small aggregates observed under the microscope, which are formed very quickly, microcolonies.

In summary, our data suggest that phototaxis in *T. vulcanus* has a wavelength dependence on green light for positive phototaxis. The cell displacement was increased from 20 to 70 µmol m⁻² s⁻¹ of green light and remained positive even at 700 µmol m⁻² s⁻¹, which is roughly one-third of the intensity of direct sunshine (*Figure 2—figure supplement 2*). Note that under nonphysiological conditions at

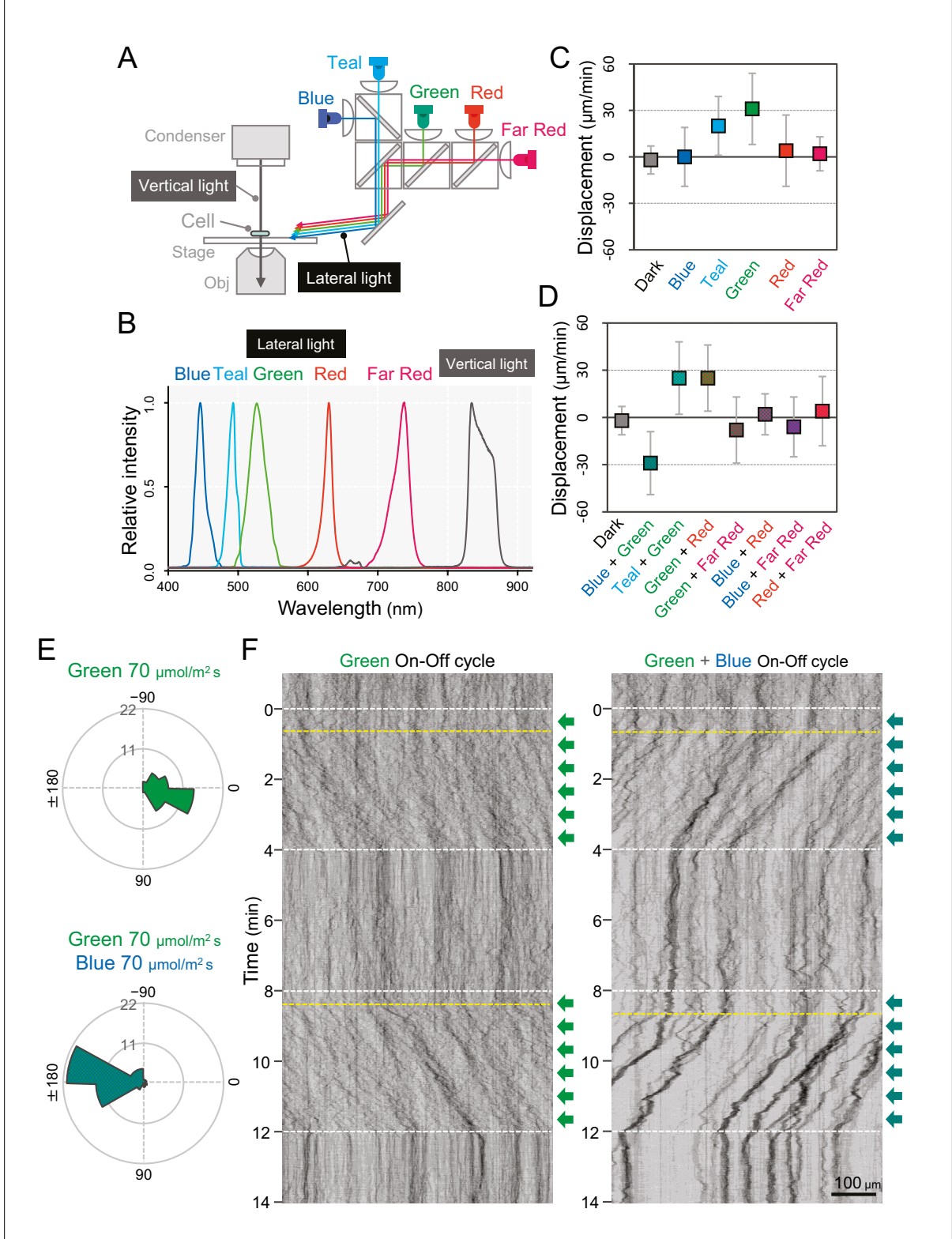

**Figure 2.** Wavelength dependency of phototaxis via optical microscopy. (**A**) Schematics of the lateral illumination for phototaxis. Five LEDs were simultaneously applied through dichroic mirrors from the right side. (**B**) Spectra of lateral and vertical light for phototaxis. (**C**) Effects of the monochromatic light source on the phototactic behaviour of cells on a glass surface. (**D**) Effects of dichromatic light source on the phototactic behaviour of cells over the glass surface. Each lateral light was used at a fluence rate of 70 μmol m$^{-2}$ s$^{-1}$. The average and standard deviation (SD) of the cell displacement along the light axis are presented ($N$ = 50). (**E**) Rose plots under green light at 70 μmol m$^{-2}$ s$^{-1}$ (upper) and green and blue light at 70 μmol

*Figure 2 continued on next page*

*Figure 2 continued*

m$^{-2}$ s$^{-1}$ (lower). The moving direction of a cell that translocated more than 6 μm min$^{-1}$ was counted. Angle 0 was the direction towards the lateral light source (*N* = 50 cells). (**F**) On–off control of phototaxis. A kymogram of cell movements along the optical axis of lateral illumination is presented. Directional movements of cells are shown by the tilted lines over time. The tilted lines from the left-upper to the right-lower side and from the right-upper to the left-lower side presented positive and negative phototaxis, respectively. Lateral illumination was applied with a time interval of 4 min and indicated by the dashed white lines (see also *Videos 5 and 6*). The delay of the cell response after the illumination was turned on is indicated by the dashed yellow lines.

The online version of this article includes the following source data and figure supplement(s) for figure 2:

**Source data 1.** Effects of the light source on the phototactic behaviour of cells on a glass surface.

**Source data 2.** Rose plots.

**Figure supplement 1.** Cell movement after applying a monochromatic light source.

**Figure supplement 2.** Cell movement at various fluence rates of a lateral green light.

**Figure supplement 3.** Phototactic behaviour of wild type (WT) cells at various temperatures.

**Figure supplement 4.** Cell movement after applying a dichromatic light source.

**Figure supplement 5.** Dose dependency of blue light to induce negative phototaxis.

**Figure supplement 6.** Effect of blue light illumination from the other side of green light on phototaxis.

room temperature (25°C), the cells showed negative phototaxis under lateral green light due to a so far unknown mechanism (*Figure 2—figure supplement 3* and *Video 3*). Therefore, we performed all further experiments at the normal growth temperature of *T. vulcanus* at 45°C using the special microscopy setup developed here (see Materials and methods).

Next, we observed cell movement with a combination of two LEDs of different wavelengths simultaneously. We found that the direction of the phototactic response was reversed after a combination of green and blue light (*Figure 2D* and *Video 4*), whereas the cells retained positive movement when green was combined with teal or red light. No directional movement was observed without green light (*Figure 2—figure supplement 4*). These data suggest that blue light is responsible for the directional switch of phototaxis. To study these effects in more detail, we applied constant green-light intensity at 70 μmol m$^{-2}$ s$^{-1}$ and combined it with different blue light intensities (*Figure 2—figure supplement 5*). Illumination with 20 μmol m$^{-2}$ s$^{-1}$ blue light in addition to green light led only to a decrease in cell displacement, whereas 200 μmol m$^{-2}$ s$^{-1}$ blue light induced negative phototaxis in all cells. At more than 700 μmol m$^{-2}$ s$^{-1}$ blue light, however, the directionality of negative phototaxis became decreased (*Figure 2—figure supplement 5E*). In contrast, microcolony formation was enhanced under these conditions (*Figure 2—figure supplement 5A*). Notably, negative phototaxis was induced even when we applied blue light from the opposite side of the lateral green-light source, suggesting that the reversal of the phototaxis from positive to negative does not depend on the direction of blue light illumination (*Figure 2—figure supplement 6*). In the following experiments, we used green light at

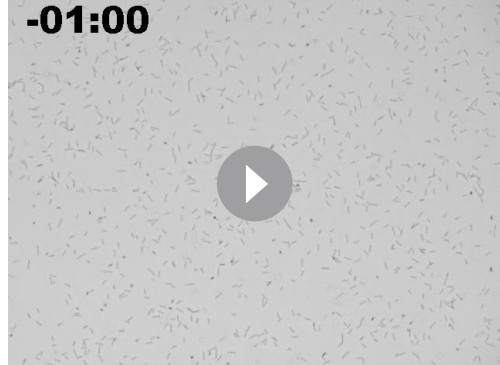

**Video 1.** Positive phototaxis. Lateral illumination of green light was turned on at time 0 from the right side of the movie. Wild type (WT) cells at 45°C. Area 750 × 563 μm.

https://elifesciences.org/articles/73405/figures#video1

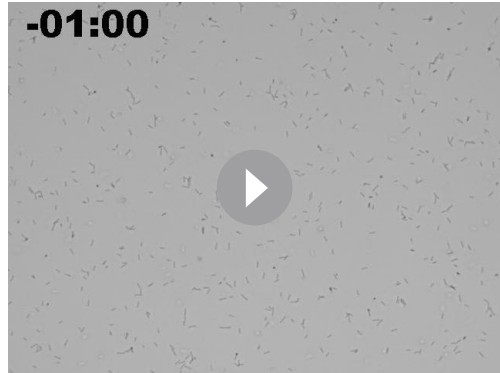

**Video 2.** Microcolony formation. Lateral illumination of blue light was turned on at time 0 from the right side of the movie. Wild type (WT) cells at 45°C. Area 750 × 563 μm.

https://elifesciences.org/articles/73405/figures#video2

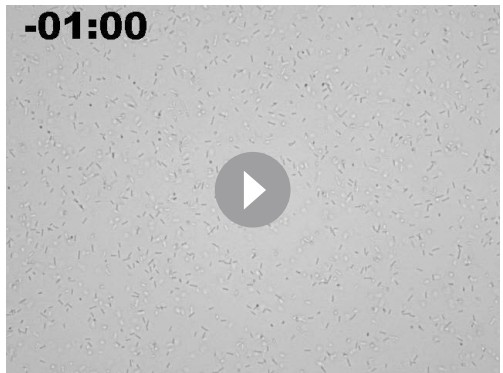

**Video 3.** Phototaxis at low temperature. Lateral illumination of green light was turned on at time 0 from the right side of the movie. Wild type (WT) cells at 25°C. Area 750 × 563 μm.

https://elifesciences.org/articles/73405/figures#video3

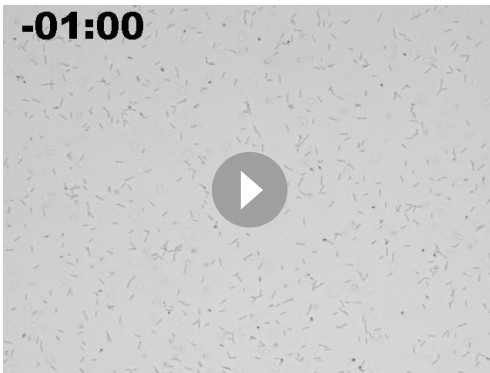

**Video 4.** Negative phototaxis. Lateral illumination of green and blue light was turned on at time 0 from the right side of the movie. Wild type (WT) cells at 45°C. Area 750 × 563 μm.

https://elifesciences.org/articles/73405/figures#video4

70 μmol m$^{-2}$ s$^{-1}$ for positive phototaxis and the combination of green light at 70 μmol m$^{-2}$ s$^{-1}$ and blue light at 200 μmol m$^{-2}$ s$^{-1}$ for negative phototaxis (*Figure 2E*). Under these conditions, the kymographs representing the spatial position of the cells along the lateral light axis over time depict examples of both positive and negative phototaxis as tilted lines (*Figure 2F*). Directed movement was barely observed for the first 30 s after the light was turned on in most cells under repeated on–off cycles of lateral illumination (*Videos 5 and 6*). This delay has also been reported in phototaxis of *Synechocystis* (*Nakane and Nishizaka, 2017*).

## Photoreceptors for phototaxis

To identify the photoreceptors involved in the phototaxis of *T. vulcanus*, we constructed mutants lacking the genes for the cyanobacteriochromes SesA (*Enomoto et al., 2014*), SesB, and SesC (*Enomoto et al., 2015*), the putative circadian input protein CikA (*Narikawa et al., 2008*), the BLUF-domain protein PixD (*Okajima et al., 2005*), the putative photoreceptor LOV (*Mandalari et al., 2013*), and the orange-carotenoid protein OCP (*Muzzopappa et al., 2017*), all of which are predicted to sense within the blue-to-green light region (*Figure 3A*). Note that no red/far-red absorbing phytochrome photoreceptor gene has been identified in the complete genome of *T. vulcanus*. All mutants showed positive phototaxis towards green light (*Figure 3B*, upper), suggesting that none of these photoreceptors is involved in the control of positive phototaxis. Next, we applied green and blue light to check negative phototaxis and measured the displacement of cell displacement along the light axis

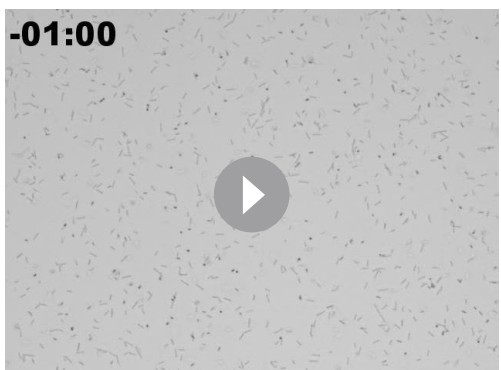

**Video 5.** On–off control of positive phototaxis. Lateral illumination of green light was applied with 4-min intervals from the right side of the movie. Wild type (WT) cells at 45°C. Area 750 × 563 μm.

https://elifesciences.org/articles/73405/figures#video5

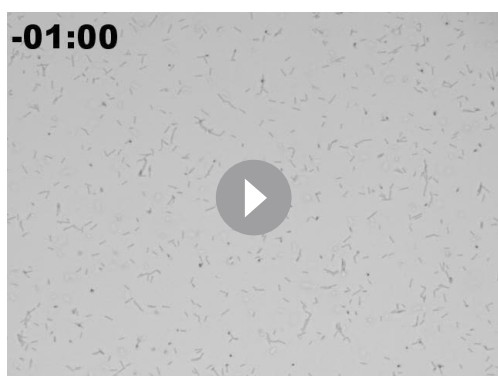

**Video 6.** On–off control of negative phototaxis. Lateral illumination of green and blue light was applied with 4-min intervals from the right side of the movie. Wild type (WT) cells at 45°C. Area 750 × 563 μm.

https://elifesciences.org/articles/73405/figures#video6

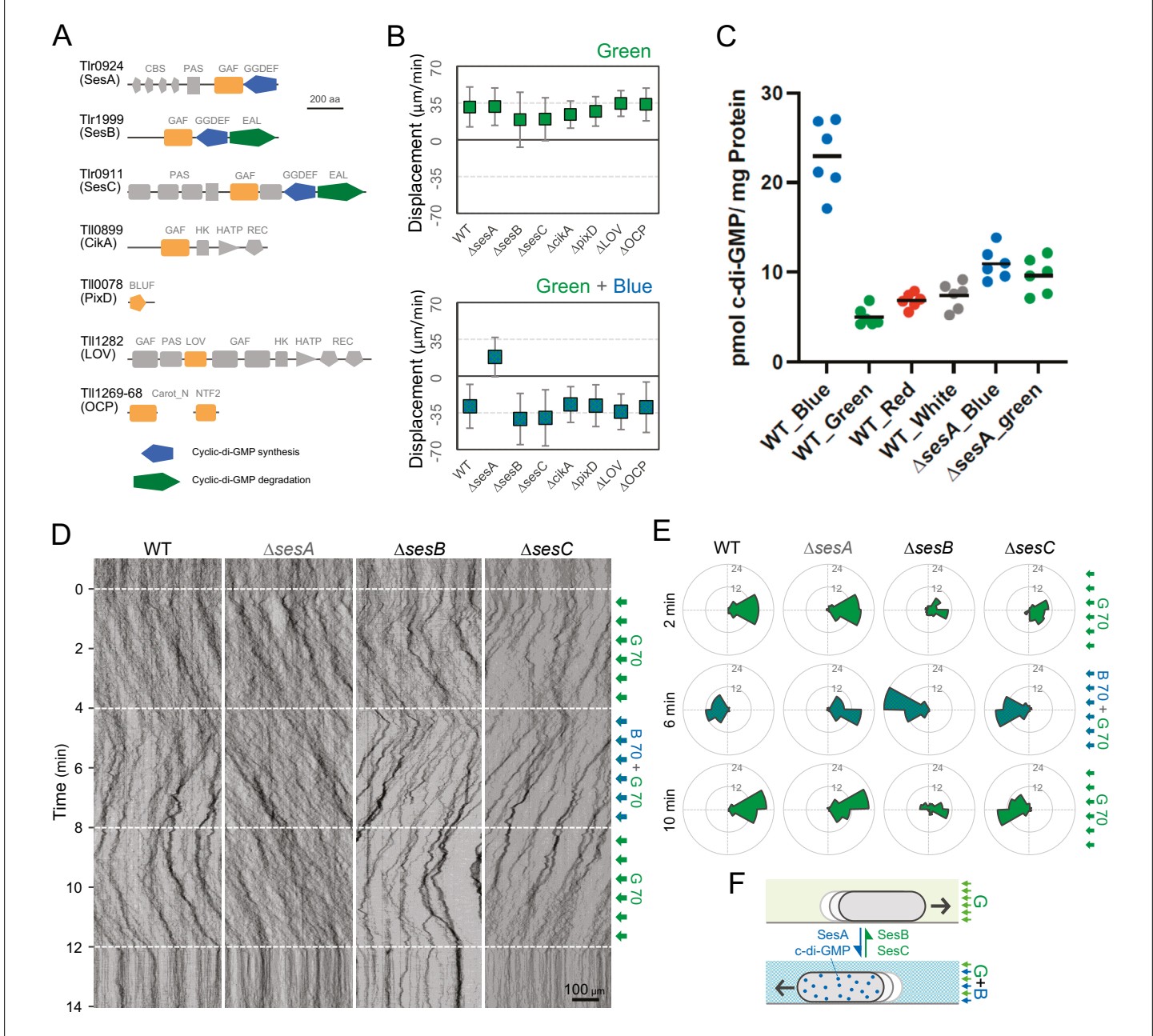

**Figure 3.** Photoreceptors for phototaxis. (**A**) Domain composition of candidate photoreceptor-containing proteins in *T. vulcanus*. (**B**) Mutant cell behaviour after lateral light illumination. *Upper*: lateral green light at a fluence rate of 70 μmol m$^{-2}$ s$^{-1}$. *Lower*: lateral green and blue light at a fluence rate of 70 μmol m$^{-2}$ s$^{-1}$. The cell displacement for a duration of 1 min was measured at 3 min after lateral illumination was turned on (*N* = 50 cells). Cell movements towards the light source are shown as a positive value. (**C**) Intracellular cyclic dimeric GMP (c-di-GMP) concentrations in wild type (WT) and Δ*sesA*. The cells were cultivated under blue, green, red, or white light illumination for 30 min, and c-di-GMP was extracted and quantified. The shown data are biological triplicates with technical duplicates, and the mean values are given with the bars. (**D**) Kymograph of cell movements in the WT, Δ*sesA*, Δ*sesB*, and Δ*sesC* mutants along the optical axis of lateral illumination. The cell position was visualized by near-infrared light. Phototaxis was stimulated by the lateral illumination of green and green/blue light. Green light was applied from time 0 to 4 and from 8 to 12 min, and green/blue light was applied from time 4 to 8 min (see also *Videos 7–10*). (**E**) Rose plot. The moving direction of a cell that translocated more than 6 μm min$^{-1}$ was counted. Lateral light was applied from the right side. The cell displacement was measured at each time point, as presented on the left (*N* = 50 cells). The data come from Panel D. (**F**) Schematic model of the reversal in phototaxis induced by the intracellular concentration of c-di-GMP.

The online version of this article includes the following source data and figure supplement(s) for figure 3:

**Source data 1.** Mutant cell behaviour after lateral light illumination.

*Figure 3 continued on next page*

*Figure 3 continued*

**Source data 2.** Rose plots.

**Source data 3.** Intracellular cyclic dimeric GMP (c-di-GMP) concentrations in wild type (WT) and Δ*sesA*.

**Figure supplement 1.** Schematic illustration of cellular cyclic dimeric GMP (c-di-GMP) levels and cell migration within a dense cyanobacterial community under solar irradiation.

(*Figure 3B*, lower). Whereas WT cells showed negative phototaxis upon lateral illumination with green and blue light, the Δ*sesA* mutant maintained positive phototaxis. SesA harbours a GGDEF domain and synthesizes the second messenger c-di-GMP in response to blue light (*Enomoto et al., 2015*). This suggests that the directional switch from positive to negative phototaxis may be triggered by SesA-dependent c-di-GMP synthesis. This assumption is supported by measurements of the intracellular c-di-GMP concentration (*Figure 3C*). The WT showed a more than three times higher c-di-GMP content under blue light than under the other tested light conditions, such as green, red, and white light. In contrast, Δ*sesA* lost the blue light-dependent increase in intracellular c-di-GMP levels.

To get further insights into the mechanism of directional switching, we have isolated a spontaneous mutant (named WT_N) that shows constitutive negative phototaxis under lateral illumination with white light. Comparative genomic analysis revealed a frameshift mutation in the gene *tll1859*, which encodes a GGDEF/EAL domain-containing protein. The frameshift mutation was found in the sequence encoding the C-terminal region of the EAL domain resulting in removal of a large part of this domain. Therefore, we expect that the mutation abolishes the c-di-GMP degradation activity of the *tll1859* gene product even in case the N-terminal part of the protein was still expressed (*Figure 4A,B*). We introduced the same frameshift mutation in the original WT background which shows positive phototaxis. The resulting mutant cells (*tll1859toN*) showed negative phototaxis on agar plates under lateral illumination with white light, similar to the spontaneous mutant WT_N (*Figure 4C*). Single-cell trajectories of *tll1859toN* cells demonstrated that they move away from green light (*Figure 4D–G*), in contrast to the WT background strain which shows positive phototaxis under these conditions (*Figure 2*). Further, we determined the intracellular c-di-GMP concentration of WT_N and *tll1859toN* cells under white light. In both strains, the c-di-GMP concentration was more than two times higher than in the original WT (*Figure 4H*). Taken together, this analysis provides independent evidence that an increase in the intracellular c-di-GMP level can control the directional switch in phototaxis.

Previous reports have shown that the three photoreceptor proteins SesA, SesB, and SesC work together for c-di-GMP-dependent control of cell aggregation via the regulation of cellulose synthesis (*Enomoto et al., 2015*; *Enomoto et al., 2014*). SesB degrades c-di-GMP, and its activity is upregulated under teal light irradiation. SesC is a bifunctional protein with enhanced c-di-GMP-producing activity under blue light and enhanced c-di-GMP-degrading activity under green light. To examine the directional switch of phototaxis in detail, we applied lateral illumination in three phases in the order of green, green/blue, and green with a time interval of 4 min (*Figure 3D,E* and *Video 7*). WT cells clearly showed directional movement and switched from positive to negative and back to positive phototaxis in response to the applied light regime. However, Δ*sesA* mutant cells maintained positive phototaxis under all conditions, even after green/blue illumination (*Figure 3D,E* and *Video 8*). In contrast, the Δ*sesC* mutant switched to negative phototaxis but did not move again towards the light source upon illumination with only green light (*Figure 3D,E* and *Video 9*). The Δ*sesB* mutant switched movement from positive to negative and from negative to positive phototaxis, similar to the WT. However, the Δ*sesB* cells continued to aggregate after the light stimulus had stopped (*Figure 3D,E* and *Video 10*), while the WT dispersed in a few minutes. We hypothesize that negative phototaxis is induced by SesA-dependent c-di-GMP synthesis under blue light, whereas SesC and SesB degrade c-di-GMP under green light, thereby inducing positive phototaxis and controlling the dispersion of microcolonies or cell aggregation (*Enomoto and Ikeuchi, 2020*; *Figure 3F*).

## Components of T4P machinery for phototaxis

T4P are known to be essential for phototactic motility in other cyanobacteria (*Bhaya et al., 2000*; *Schuergers et al., 2014*; *Yoshihara et al., 2001*). To evaluate the contribution of T4P to *T. vulcanus* phototaxis, we tested the phenotype of Δ*pilA1*, Δ*pilB*, Δ*pilT1*, and Δ*hfq* mutants (*Figure 5A*). Electron microscopy revealed that the Δ*pilB* and Δ*hfq* mutant cells lacked pilus filaments, whereas WT cells

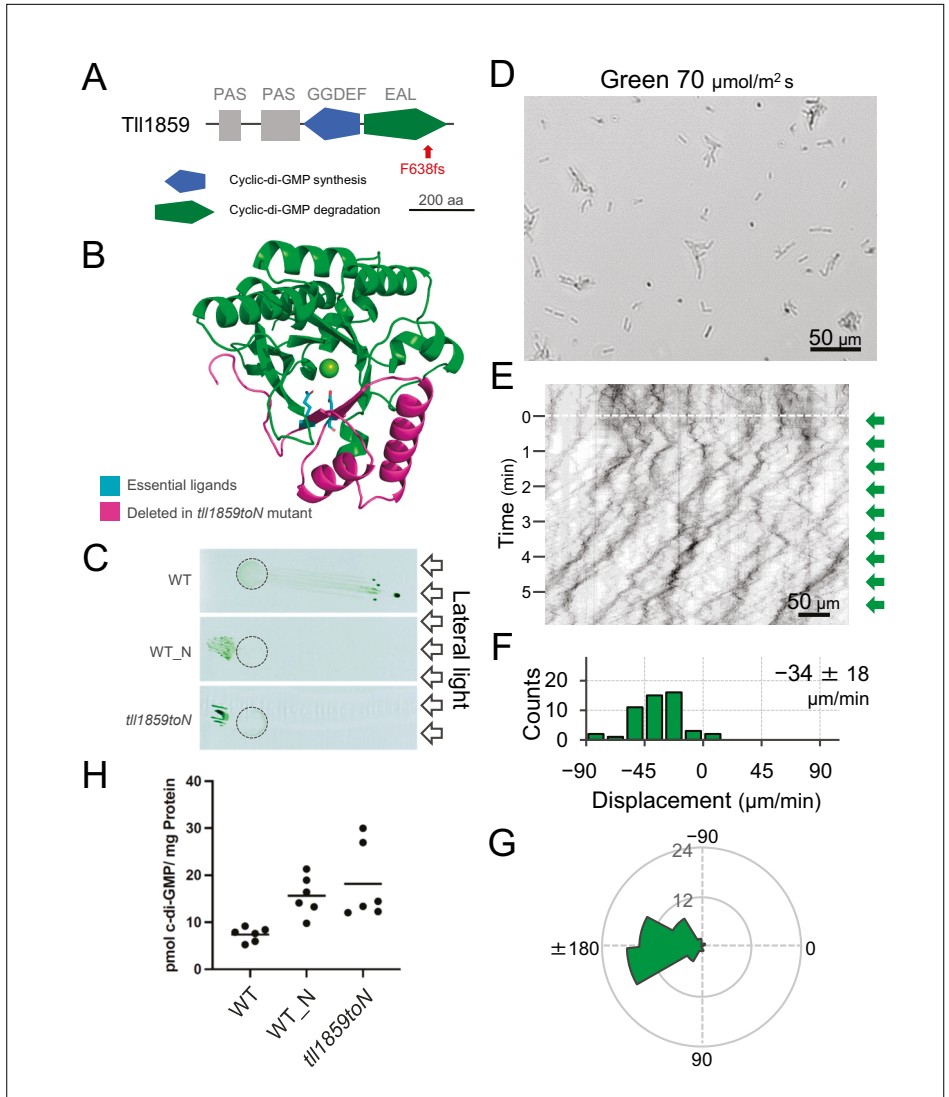

**Figure 4.** Effect of the *tll1859* mutation on phototaxis and intracellular cyclic dimeric GMP (c-di-GMP) concentrations. (**A**) Domain composition of Tll1859 protein. The WT_N strain has a frameshift mutation in the gene sequence of the EAL domain. (**B**) Predicted structure of the Tll1859-EAL domain by SWISS-MODEL. The deleted region in WT_N strain is coloured by magenta. The putative ligand residues essential for metal ion binding (Glu669 and Gln689) which are absent in WT_N, are highlighted in cyan. (**C**) Phototaxis on agar plates under lateral illumination of white light at a fluence rate of 150 μmol m$^{-2}$ s$^{-1}$. (**D**) Intracellular c-di-GMP concentrations in WT, WT_N, and the independently created *tll1859toN* mutant. The shown data are biological triplicates with technical duplicates, and the mean values are given with the bars. (**E**) Cell images at 4 min after the lateral green light was turned on. (**F**) Kymograph of cell movements along the optical axis of lateral illumination. Directional movements of cells are presented by the tilted lines over time. Lateral green-light illumination was turned on at time 0, presented as a dashed white line. (**G**) Histograms of the cell displacement along the lateral light axis. Cell movements towards the light source are shown as a positive value. The average and standard deviation (SD) of the cell displacement along the light axis are presented (*N* = 50). (**H**) Rose plots. The moving direction of a cell that translocated more than 6 μm min$^{-1}$ was counted. Angle 0 was the direction towards the lateral light source of the green light. The cell displacement for a duration of 1 min was measured at 4 min after lateral illumination was turned on (*N* = 50 cells).

The online version of this article includes the following source data for figure 4:

**Source data 1.** Intracellular cyclic dimeric GMP (c-di-GMP) concentrations in WT_N and *tll1859toN* mutant.

**Source data 2.** List of nucleotide differences found in 'WT_N' compared with 'WT' of *T*.

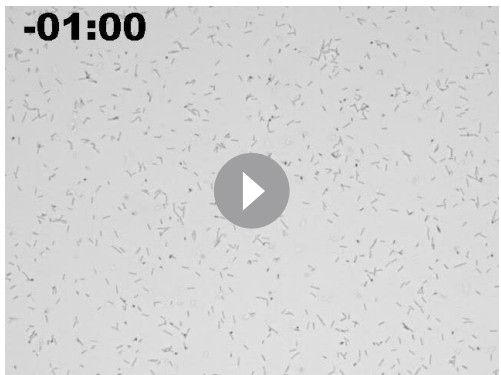

**Video 7.** Directional switch of phototaxis in wild type (WT). Lateral illumination was turned on at time 0 from the right side of the movie. The illumination was applied with three phases in the order of green, green/blue, and green with a time interval of 4 min. WT cells at 45°C. Area 750 × 563 µm.

https://elifesciences.org/articles/73405/figures#video7

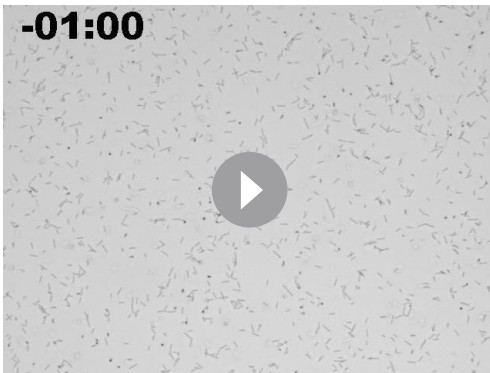

**Video 8.** Cell behaviour in ΔsesA mutant. Lateral illumination was turned on at time 0 from the right side of the movie. The illumination was applied with three phases in the order of green, green/blue, and green with a time interval of 4 min. ΔsesA cells. Area 750 × 563 µm.

https://elifesciences.org/articles/73405/figures#video8

had dozens of filaments localized at each cell pole (*Figure 5B,C*). As expected, the mutants did not show phototaxis upon green-light illumination under the current microscope setup (*Figure 5D* and *Video 11*). These data showed that T4P filaments are polarly localized in *T. vulcanus*, which is necessary to generate directional movement in response to light illumination.

## Random orientation of cells during negative phototaxis

We observed that cells formed more microcolonies during negative phototaxis than during positive phototaxis (*Videos 1 and 4*). The cells started to aggregate 1 min after the green and blue light stimulus (*Video 4*) but nevertheless kept the directed movement away from the light source (*Figure 6A*, upper and *Video 12*). Thus, microcolony formation of cells did not hinder directed movements. Under these conditions, approximately 70% of all cells formed such microcolonies and moved directionally, while 30% of cells stayed as single cells (*Figure 6B*). These cells were able to adopt a random orientation along the light vector, meaning that they aligned their axis at different angles in relation to the light direction and exhibited directed movements as single cells. As an example, we show one cell

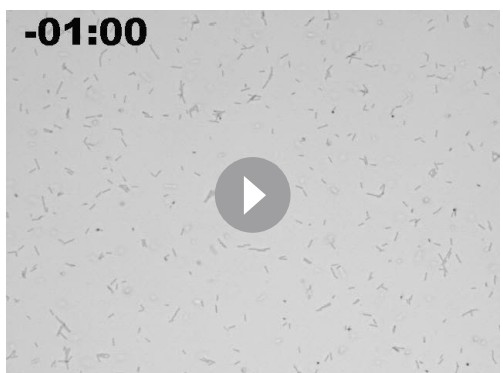

**Video 9.** Cell behaviour in ΔsesC mutant. Lateral illumination was turned on at time 0 from the right side of the movie. The illumination was applied with three phases in the order of green, green/blue, and green with a time interval of 4 min. ΔsesC cells. Area 750 × 563 µm.

https://elifesciences.org/articles/73405/figures#video9

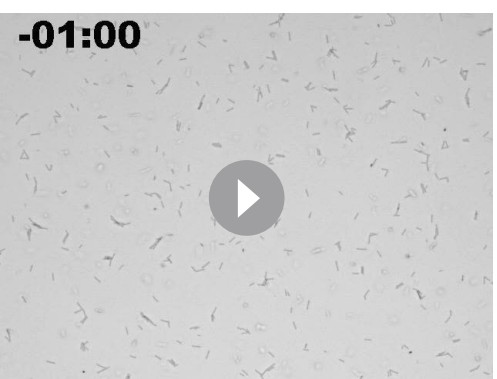

**Video 10.** Cell behaviour in ΔsesB mutant. Lateral illumination was turned on at time 0 from the right side of the movie. The illumination was applied with three phases in the order of green, green/blue, and green with a time interval of 4 min. ΔsesB cells. Area 750 × 563 µm.

https://elifesciences.org/articles/73405/figures#video10

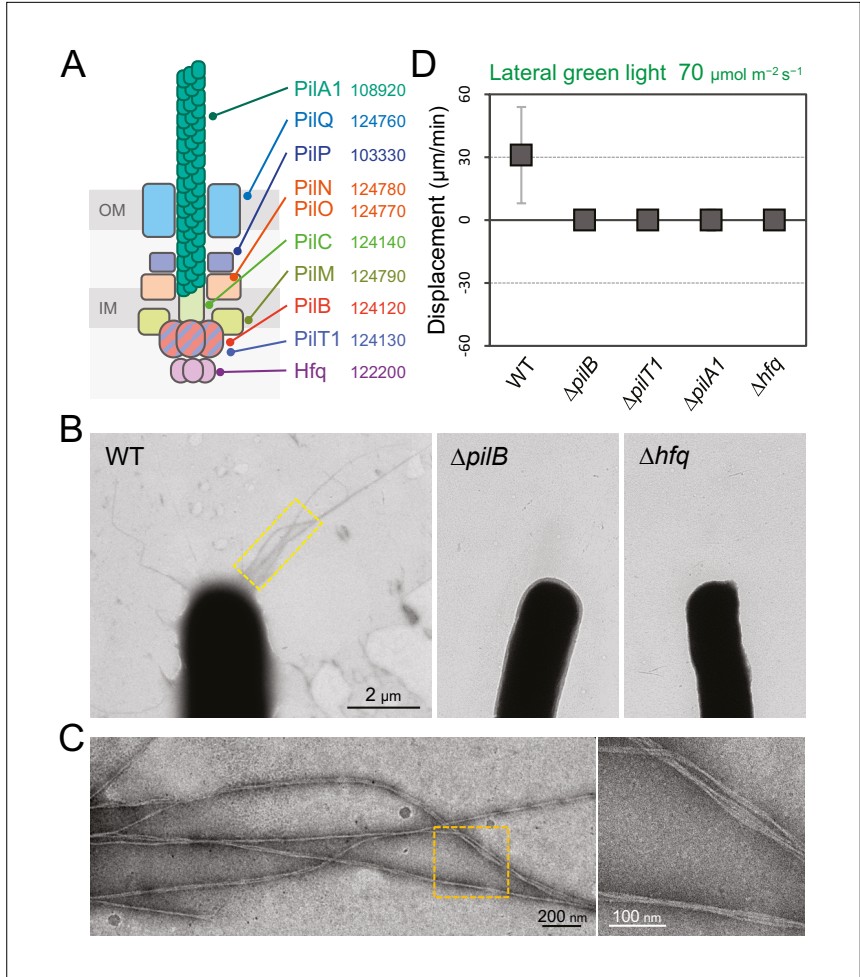

**Figure 5.** Type IV pili (T4P) machinery and phototaxis. (**A**) Schematic of the T4P machinery. Protein components and their gene IDs from the genome of *T. vulcanus* are indicated. The location of the components is presented in reference to other species (*Chang et al., 2016*). (**B**) Electron microscopy (EM) image of a cell. (**C**) Magnified images of the yellow and orange boxed areas are presented. (**D**) Effect of lateral light illumination on mutant cells. Lateral green light was applied at a fluence rate of 70 µmol m⁻² s⁻¹. The average and standard deviation (SD) of the cell displacement on the glass surface along the light axis are presented (*N* = 50). Hfq was previously suggested to be involved in the formation of T4P machinery in other cyanobacteria (*Schuergers et al., 2014*).

The online version of this article includes the following source data for figure 5:

**Source data 1.** Effect of lateral light illumination on mutant cells.

aligned perpendicular to the optical axis of lateral light and attached at both poles to the surface (*Figure 6A*, left bottom and *Video 13*). Another example is a cell positioned in an upright position at one pole (*Figure 6A*, right bottom and *Video 14*), reminiscent of walking observed previously in *Pseudomonas* (*Gibiansky, 2010*). Not all single cells aligned in parallel along the light axis (*Figure 6C*), suggesting that the force for cell movement is not generated exclusively along the longer axis of the cell.

## Direct visualization of T4P dynamics using fluorescent beads

We visualized the T4P dynamics of *T. vulcanus* through beads attached to the pilus fibre (*Figure 7A* and *Video 15*), as previously described in a coccoid-shaped unicellular cyanobacterium (*Nakane and Nishizaka, 2017*). In the presence of 200-nm diameter polystyrene sulfate beads, the beads around the cell pole showed directional movements (*Figure 7B,C*). The distance between beads and the cell pole increased or decreased with time (*Figure 7D*). The bead movement was biased to the cell poles, resulting in the accumulation of beads at the poles after a few minutes of observation (*Figure 7—figure*

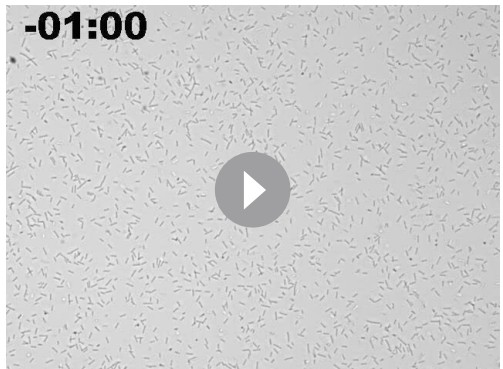

**Video 11.** Cell behaviour of a Δ*pilB* mutant. Lateral illumination of green light was turned on at time 0 from the right side of the movie. Δ*pilB* cells at 45°C. Area 750 × 563 μm.
https://elifesciences.org/articles/73405/figures#video11

*supplement 1A*). This accumulation was not observed in the Δ*pilB* mutant (*Figure 7—figure supplement 1A*), suggesting that directed bead movement is driven by dynamic T4P activity. The average velocity of beads towards and away from cell poles was measured to be 1.59 ± 0.34 and 2.68 ± 0.33 μm s$^{-1}$ (*Figure 7E*), corresponding to the retraction and extension of T4P, respectively (*Nakane and Nishizaka, 2017*). The distribution of the velocity was not changed during illumination with green and green/blue light (*Figure 7—figure supplement 2*). Note that the sulfate beads were effectively captured by T4P, whereas neither carboxylate- nor amine-modified beads accumulated (*Figure 7—figure supplement 1B*), suggesting that the surface modification is crucial for specific binding to the T4P filament. As we did not detect filaments that were covered by the beads, we suggest that they only bind to a specific part of the filament, which is most likely the tip of the pilus.

We also visualized the T4P dynamics through beads during negative phototaxis (*Figure 7F* and *Video 16*). Here, only cells that were aligned perpendicular to the optical axis of lateral light were used for data analyses. The beads were retracted only at the cell pole facing the opposite side of the light source, which is the front side of the cell in negative phototaxis (*Figure 7G*). This suggests that the T4P are asymmetrically activated within a single-cell pole (*Figure 7H*).

## Direct visualization of T4P dynamics by fluorescent labelling of PilA1

To observe T4P filaments specifically in live cells, we introduced a cysteine residue into the PilA1 subunit to allow for labelling with maleimide-conjugated fluorophore (*Ellison et al., 2017*). For site-directed mutagenesis, we selected the codon for serine at position 114 in *pilA1* and introduced the respective mutation in *Thermosynechococcus*. This residue was chosen because the homologous position (T126) of the MshA subunit was successfully used in *V. cholerae* (*Ellison et al., 2019*). After labelling, we directly observed extension and retraction of pilus filaments of the PilA1$^{S114C}$ mutant on a glass surface by epi-fluorescence microscopy (*Figure 8A*). The T4P filaments were localized at both cell poles and dynamic movement of pili towards and away from the cell poles was observed (*Video 17*). No pili were detected in the middle of the cells, supporting our previous measurements which showed T4P localization only at both poles. The average velocity of T4P filaments was measured to be 1.6 ± 0.4 and 2.3 ± 0.6 μm s$^{-1}$ for extension and retraction, respectively (*Figure 8B*). Although the light intensity for the excitation of the fluorescence dye was much higher than in the experiments of *Figures 2 and 3*, the T4P dynamics by the fluorescent labelling of PilA1 had a similar rate as measured in the beads assay (*Figure 7E*). When the cell showed directional movement perpendicular to its longer axis, bipolar T4P dynamics were asymmetric and observed at the front side of the rod in the direction of movement (*Figure 8C* and *Video 18*). This was also confirmed by the observations using total internal reflection fluorescence (TIRF) microscopy, which enable to excite the fluorescence dye near the glass surface (*Figure 8D* and *Video 19*). Pilus filaments appeared asymmetrically at a few micrometres distance from both cell poles in the direction of cell movement. Most probably, TIRF microscopy revealed those pili which adhered to the glass surface for cellular movement (*Figure 8E,F* and *Video 20*). These data evidently indicate that the asymmetric activation of T4P within a cell pole guides the cellular movement via extension–attachment–retraction cycles in a certain direction.

## Discussion

Here, we demonstrated that directional movement in phototaxis can be reversed by illumination with lateral light of another wavelength on a short time scale (*Figures 2 and 3*). In bacterial twitching

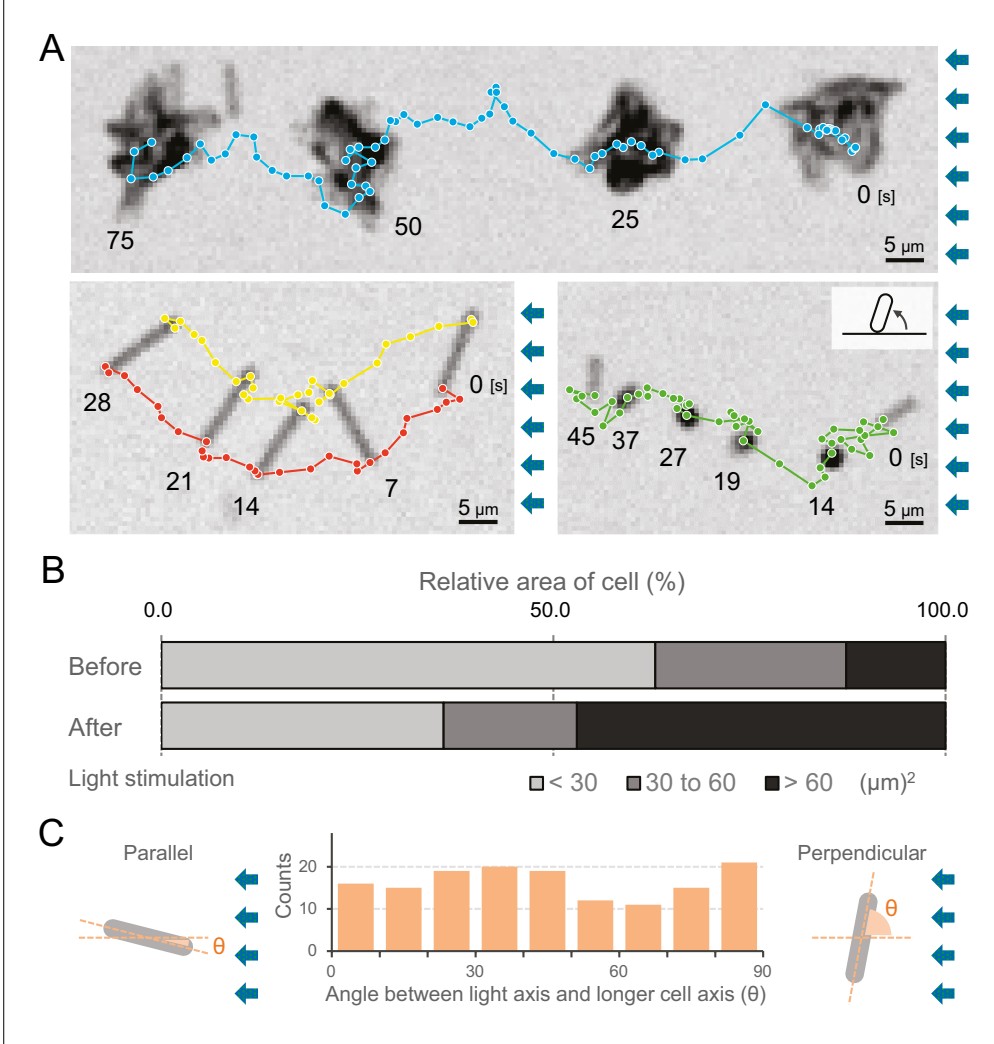

**Figure 6.** Moving trajectories of cells during negative phototaxis. (**A**) Cell images and moving trajectories. Images were integrated with a single image at each time duration presented (see also *Videos 12–14*). Upper: movement of microcolonies. Left bottom: cell perpendicular to lateral light axis. Right bottom: cell that stood up and kept binding at a cell pole. (**B**) Cell–cell interaction. The area of a cell moving as a single unit was measured before and after the induction of negative phototaxis and presented as the ratio by the area (N = 353). (**C**) Moving direction of a cell in relation to the light source. The orientation of single cells during negative phototaxis was measured, and the distribution is presented (N = 148). The absolute angle of the longer axis of a cell was measured in relation to the lateral light axis. The parallel cellular orientation to the light axis (shown on the left) was taken as 0°, whereas the perpendicular orientation (shown on the right) was ideally 90°.

The online version of this article includes the following source data for figure 6:

**Source data 1.** Moving trajectories.

**Source data 2.** Area of a cell.

**Source data 3.** Moving direction of a cell in relation to the light source.

motility, cellular movement is driven by repeated cycles of extension and retraction of T4P (*Figures 5, 7 and 8*). In rod-shaped bacteria, such as *Myxococcus*, the T4P machinery is localized at cell poles and generates a directional bias presumably by the force generated at the leading pole along the longer axis of the cell (*Mercier et al., 2020*; *Potapova et al., 2020*; *Sabass et al., 2017*). Here, we show that rod-shaped *T. vulcanus* cells showed directional movement even if the cell orientation was perpendicular to the lateral light axis (*Figure 6*). This behaviour can be explained by asymmetric activation of T4P for the long cell axis at the pole on one side of the rod (*Figures 7 and 8*). Such localization of

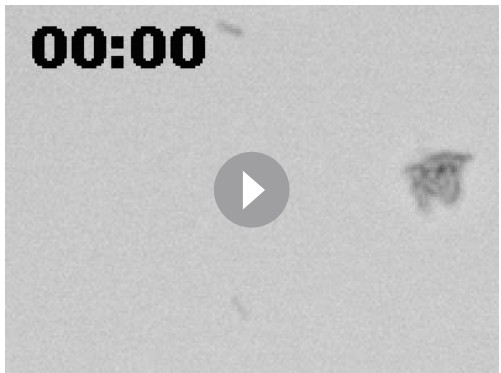

**Video 12.** Phototaxis of microcolony. Lateral illumination of green and blue light was applied from the right side of the movie. Wild type (WT) cells at 45°C. Area 117 × 88 μm.
https://elifesciences.org/articles/73405/figures#video12

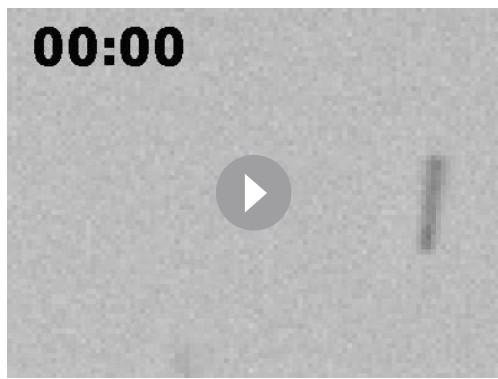

**Video 13.** Phototaxis of cell perpendicular to the lateral light axis. Lateral illumination of green and blue light was applied from the right side of the movie. Wild type (WT) cell at 45°C. Area 171 × 44 μm.
https://elifesciences.org/articles/73405/figures#video13

pili was supported by our bead assay (*Figure 7F–H*), which showed localized retraction of beads on the cell side that did not face the light source in negative phototaxis, and by the direct visualization of PilA1 filaments (*Figure 8*). Local activation of T4P along the shorter axis of the cell would require a specific signalling system with high spatial resolution compared to the well-known pole-to-pole regulation by *Myxococcus* and *Pseudomonas* (*Kühn et al., 2021*; *Mercier et al., 2020*; *Potapova et al., 2020*; *Sabass et al., 2017*). In these bacteria, a dynamic change in the localization of the motor protein PilB between both poles leads to a directional switch in motility (*Kühn et al., 2021*; *Mercier et al., 2020*). The localization of PilB also seems to govern the direction of movement in *Synechocystis* phototaxis (*Schuergers et al., 2015*). Cell polarity for the short axis of the *T. vulcanus* cells might result in localization of the PilB protein to the leading side of the cell, driving directional movement even if the cell is oriented lengthwise with respect to the light source.

It remains an open question what the photoreceptor for positive phototaxis is. We showed that the gene disruption mutants of all photoreceptors identified to date for blue-to-green light still exhibited positive phototaxis (*Figure 3*), suggesting that another unrecognized photoreceptor for positive phototaxis might remain to be identified in this species. In a wild isolate of *S. elongatus*, the cyanobacteriochrome PixJ was reported to be responsible for directional light sensing in phototaxis (*Yang et al., 2018*). A homologue of the PixJ protein was also shown to localize at cell poles in a closely related *Thermosynechococcus* species (*Kondou et al., 2002*). However, the *pixJ* orthologue is deleted from the genome of our WT strain (*Cheng et al., 2020*). Recent work reported that the GAF domain-containing photoreceptor PixJ is not responsible for the phototactic motility of the filament-forming cyanobacterium *Nostoc punctiforme* (*Harwood et al., 2021*). The authors hypothesized that the localized proton motive force is the first input for directional light sensing (*Harwood et al., 2021*). Considering the micro-optic effects in rod-shaped cyanobacteria (*Figure 7—figure supplement 3*; *Yang et al., 2018*), the spatial difference in light intensity between cell poles or different sides of the cell may be a universal key feature of T4P-dependent phototaxis. The focused green light would excite yet unknown photosensory molecules to induce spatially localized signalling, whereas the position of the focused blue light is not crucial for directional switching. As we showed, the direction of blue light illumination did not influence directionality of movement,

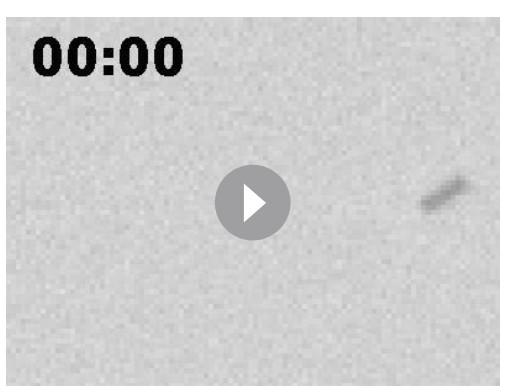

**Video 14.** Phototaxis of cell that stood up and kept binding at a cell pole. Lateral illumination of green and blue light was applied from the right side of the movie. Wild type (WT) cell at 45°C. Area 171 × 44 μm.
https://elifesciences.org/articles/73405/figures#video14

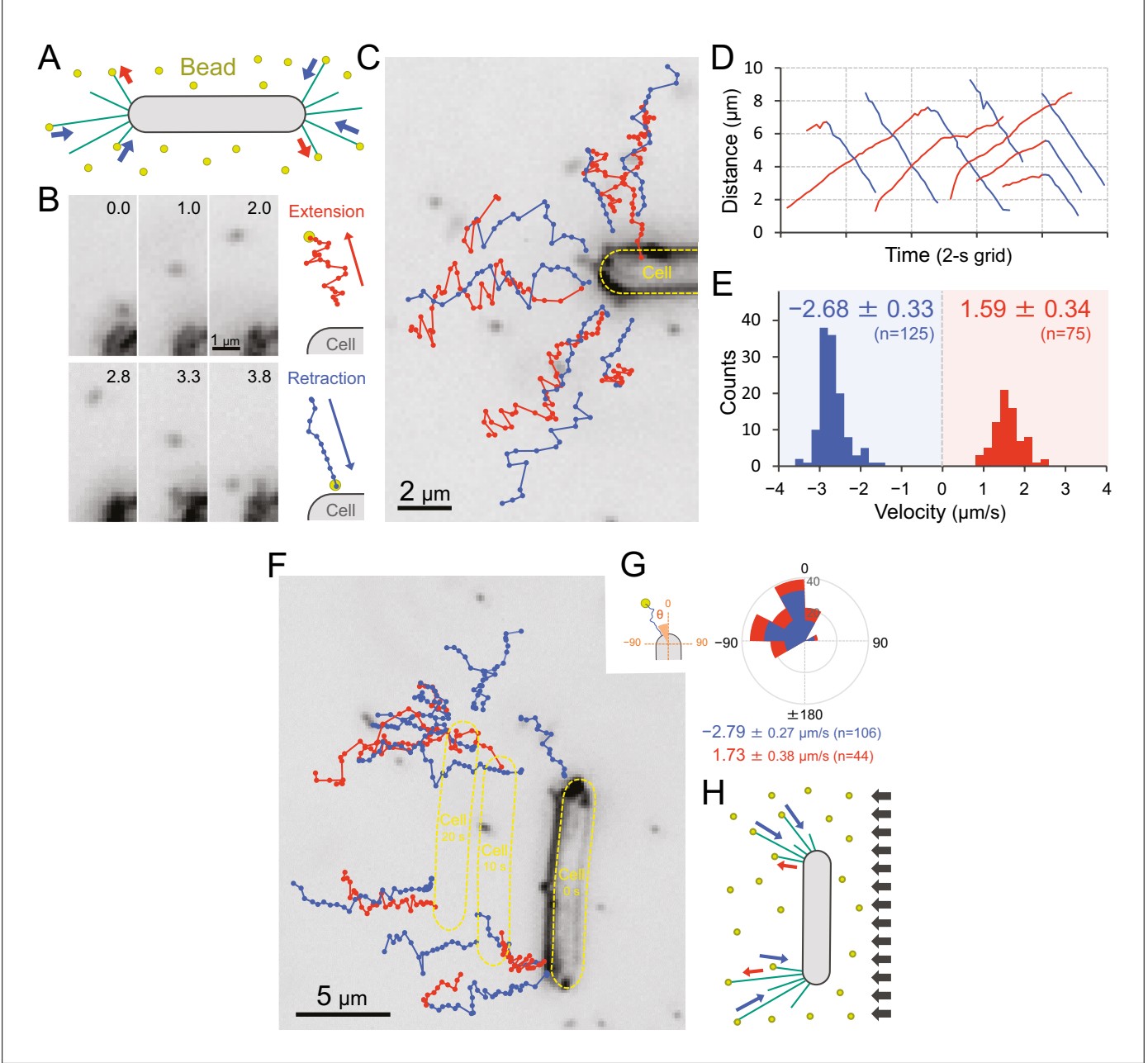

**Figure 7.** Visualization of type IV pili (T4P) dynamics through nano-beads. (**A**) Schematic of the beads' assay. Sulfate beads (200 nm in diameter) were added to the cells on a glass surface. (**B**) Typical trajectories of beads. *Left*: a series of images. Time was presented at the right upper corner of the image. *Right*: trajectories of the bead moving away from and towards the cell. (**C**) Trajectories of beads. Red and blue indicate bead movement away from and towards the cell, respectively. The bead trajectories with a time interval of 0.1 s were overlaid onto the cell image. (**D**) Time course of the distance between the bead and a cell pole. The data came from Panel C. (**E**) Distribution of the bead velocity. The velocity was measured by the time course of bead displacement. The movement towards the cell was measured as a negative value. The average and standard deviation (SD) of the plus and minus regions are presented (*N* = 200 in 12 cells). (**F**) Trajectories of beads during negative phototaxis. Lateral light illumination was applied from the right side of the image. The cell perpendicular to the lateral light axis is presented. The cell showed directional movement towards the left side of the image, and the cell position is presented by the dashed yellow lines every 10 s. The trajectories of beads with a time interval of 0.1 s were overlaid onto the cell image. (**G**) Distribution of the bead angle. *Left*: schematic of the angle definition. The cells perpendicular to the lateral light axis were used for analysis. *Right*: the angle formed by the longer cell axis and the bead trajectories around the cell pole at the upper side were measured (*N* = 150 in 10 cells). (**H**) Schematic of T4P dynamics during negative phototaxis of the cell perpendicular to the lateral light axis. T4P was asymmetrically activated on the other side of the light source.

The online version of this article includes the following source data and figure supplement(s) for figure 7:

*Figure 7 continued on next page*

*Figure 7 continued*

**Source data 1.** Moving trajectories of beads in *Figure 7BF*.

**Source data 2.** Time course of the distance between the bead and a cell pole.

**Source data 3.** Distribution of the bead velocity in *Figure 7E*, *Figure 7—figure supplement 2* –.

**Source data 4.** Rose plot.

**Figure supplement 1.** Optimization of beads assay.

**Figure supplement 2.** Effect of the illumination of green and green blue light on type IV pili (T4P) dynamics.

**Figure supplement 3.** Micro-optics effect.

because cells do not move in random orientation (*Figure 2—figure supplement 6*). Thus, blue light does not control the directional light-sensing capability, instead it provides the signal for the switch between positive and negative phototaxis. This is very similar to the situation in *Synechocystis* where the blue light receptor PixD controls the switch between negative and positive phototaxis independently of the position of the blue-light source (*Sugimoto et al., 2017*).

How could c-di-GMP regulate the motility direction of *T. vulcanus* phototaxis? The 30–60 s delay of the cellular response upon on/off illumination with blue light under background green light might result from the modulation of intracellular c-di-GMP levels or the activation of other downstream events. Considering the fast response of the directional switching of *T. vulcanus* motility in one minute, we assume that toggle regulation does not involve transcriptional regulation (*Song et al., 2011*). As additional blue illumination has the same effect on phototaxis irrespective of the position of the blue light source (*Figure 2—figure supplement 6*), we also assume that the increased global cellular pool of c-di-GMP under these conditions regulates the switch in movement.

In *Synechocystis*, multiple signalling networks seem to be integrated via PATAN-REC proteins, which interact with PilB1 (*Han et al., 2022*; *Jakob et al., 2020*). We have not yet assessed the localization of the PilB protein, but we observed that the active pilus filaments face the direction of cell movement (*Figures 7 and 8*). Notably, the MshEN domain in cyanobacterial PilB proteins is a potential c-di-GMP binding domain despite the lack of experimental evidence (*Wang et al., 2016*). The switching between negative and positive phototaxis could be mediated by the binding and unbinding of c-di-GMP to PilB, respectively. Since we did not observe a change in pilus dynamics under green and green/blue light illumination (*Figure 7—figure supplement 2*), the T4P regulation in *T. vulcanus* may not be explained simply by a specific activation of PilB (*Floyd et al., 2020*; *Hendrick et al., 2017*). Therefore, we hypothesize that c-di-GMP binding to PilB could modify the affinity of its binding to other interaction partner(s), leading to the different localization of PilB regarding the incident light vector. Blue light increases the intracellular c-di-GMP content and represses motility in *Synechocystis* on phototaxis plates (*Savakis et al., 2012*), although short-term effects have not yet been

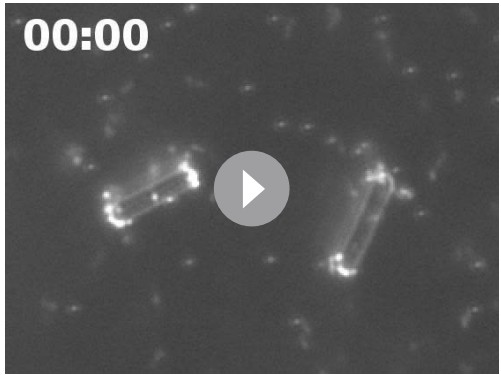

**Video 15.** Type IV pili (T4P) dynamics through beads. Extension and retraction of the T4P filament were visualized by sulfate beads with a size of 200 nm. Wild type (WT) cells at 45°C. Area 42.0 × 31.5 μm.
https://elifesciences.org/articles/73405/figures#video15

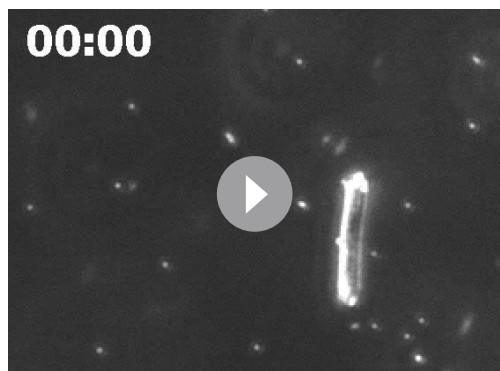

**Video 16.** Type IV pili (T4P) dynamics through beads during negative phototaxis. Lateral illumination was applied from the right side of the movie. Extension and retraction of the T4P filament were visualized by beads with a size of 200 nm. Dark-field microscopy. Wild type (WT) cell at 45°C. Area 42.0 × 31.5 μm.
https://elifesciences.org/articles/73405/figures#video16

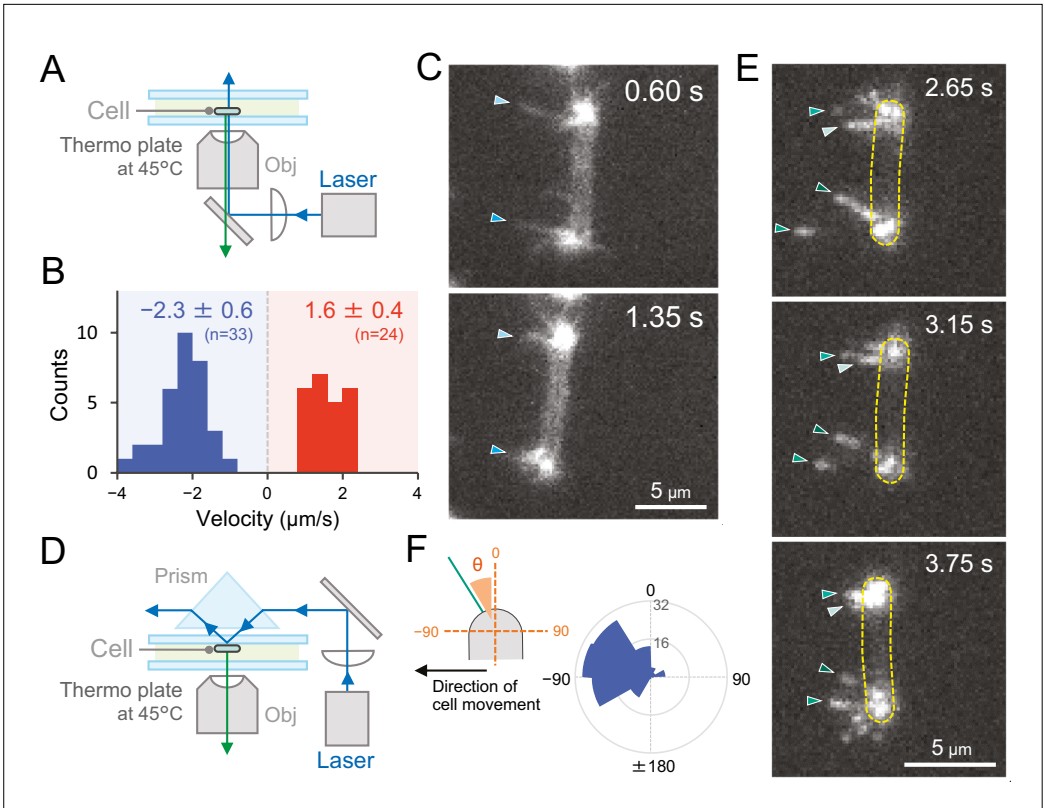

**Figure 8.** Direct visualization of type IV pili (T4P) dynamics by PilA labelling in living cells. (**A**) Schematic of the epi-fluorescent microscopy setup. (**B**) Distribution of the velocity of T4P dynamics. The velocity of pilus dynamics was measured as the time course of the displacement of the identified tip part of a T4P filament towards or away from a cell pole. The retraction of T4P was measured as a negative value. The average and standard deviation (SD) of the plus and minus values are presented ($N$ = 57 in 10 cells). The non-moving cells under epi-fluorescence microscopy were used for the data analysis. (**C**) Consecutive time-lapse images of a cell moving towards the left side under epi-fluorescence microscopy. The time is shown at the right upper corner of the image. (**D**) Schematic of the total internal reflection fluorescence (TIRF) microscopy setup. (**E**) Image sequence of a cell moving towards the left side under TIRF microscopy. The time frame is presented at the right upper corner of the image. (**F**) Distribution of the T4P filaments. *Left*: schematic of the angle definition. The cells moving towards the left side aligned perpendicular to the x axis of the image under TIRF microscopy were used for the data analysis. The angle relative to the cell axis in the direction of movement was measured as a negative value. The angle formed by the longer cell axis and the tip of T4P filaments around the cell pole were measured ($N$ = 119 in 20 cell poles).

The online version of this article includes the following source data for figure 8:

**Source data 1.** Distribution of the velocity of type IV pili (T4P) dynamics in *Figure 8B*.

**Source data 2.** Rose plot of the type IV pili (T4P) filaments in *Figure 8F*.

explored. Whether c-di-GMP could induce a similar directional switching in other cyanobacterial species remains to be clarified. Notably, *Synechocystis* also switches from positive to negative phototaxis on a relatively short time scale by a different mechanism (*Jakob et al., 2020*). High-intensity blue light is sensed by the BLUF-domain PixD photoreceptor. Upon excitation, the binding of PixD to the specific PATAN-domain CheY-like response regulator (PixE) is inhibited, and free PixE binds to PilB1, inducing a switch from red light-dependent positive to negative phototaxis (*Jakob et al., 2020*). In *T. vulcanus*, the lack of PixD did not lead to directional switching (*Figure 3B*). Thus, we suppose that different cyanobacteria have evolved specific photoreceptors and response mechanisms to control the switch between negative and positive phototaxis, presumably depending on their natural habitat and the ecophysiological context.

*T. vulcanus* was initially isolated from a mat in a hot spring in Japan (*Katoh et al., 2001*). We previously reported that in multicellular cyanobacterial communities, the ratio between blue- and

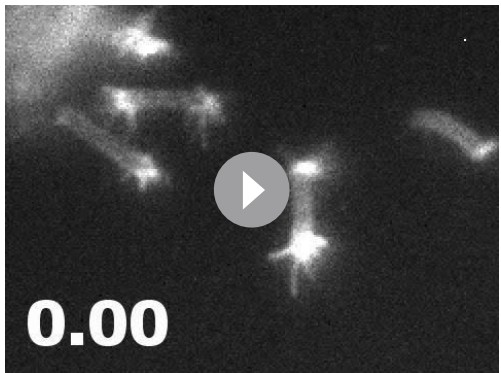

**Video 17.** Type IV pili (T4P) dynamics under epi-fluorescence microscopy. Extension and retraction of the T4P filaments were visualized by fluorescence labelling of PilA1 at 45°C. Area 39.0 × 29.3 μm.
https://elifesciences.org/articles/73405/figures#video17

**Video 18.** Type IV pili (T4P) dynamics during directional movement under epi-fluorescence microscopy. Extension and retraction of the T4P filaments were visualized by fluorescence labelling of PilA1. The cell showed directional movement towards the left side at 45°C. Area 26.0 × 19.5 μm.
https://elifesciences.org/articles/73405/figures#video18

green-light changes from surface (blue rich) to green rich within a culture due to pigment absorption. Thus, through the concerted action of the SesABC photoreceptor system, c-di-GMP production would be activated at the surface, whereas within a cyanobacterial community, low cellular c-di-GMP levels would predominate (*Enomoto and Ikeuchi, 2020*). The current study suggests that in a biofilm or a mat, cells move away from the blue light-rich surface area, whereas they move upwards under green-rich conditions within the community (*Figure 3—figure supplement 1*). Such behaviour would lead to a dynamic circulation of *Thermosynechococcus* species inside a microbial mat under solar irradiance (*Conradi et al., 2020*). The high light/short wavelength-induced downward migration and green-light-induced upward migration of cyanobacteria in microbial mats are well documented (*Bebout and Garcia-Pichel, 1995*; *Kruschel and Castenholz, 1998*; *Lichtenberg et al., 2020*; *Nadeau et al., 1999*). Future work on spatiotemporal processes in phototrophic mats can provide insights into the yet unknown molecular mechanisms of photomovements in ecophysiologically relevant niches.

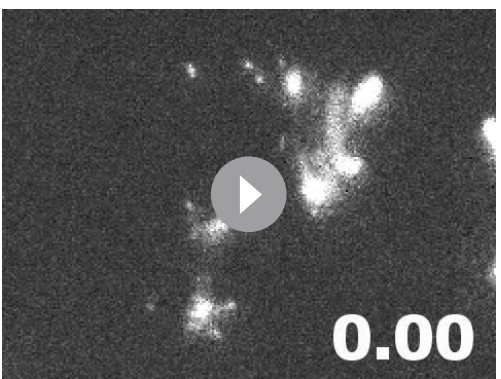

**Video 19.** Type IV pili (T4P) dynamics under total internal reflection fluorescence (TIRF) microscopy. Surface attachment of the T4P filaments was visualized by fluorescence labelling of PilA1 under TIRF illumination at 45°C. Area 39.0 × 29.3 μm.
https://elifesciences.org/articles/73405/figures#video19

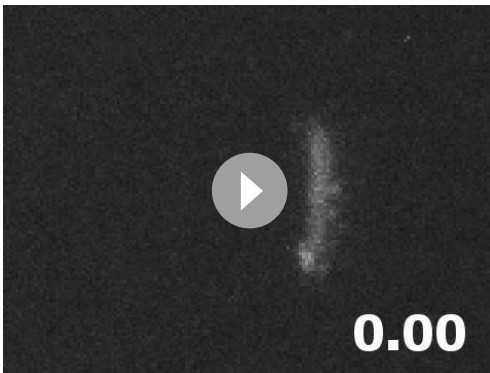

**Video 20.** Type IV pili (T4P) dynamics during directional movement under total internal reflection fluorescence (TIRF) microscopy. Surface attachment of the T4P filaments was visualized by fluorescence labelling of PilA1 under TIRF illumination. The cell showed directional movement towards the left side at 45°C. Area 26.0 × 19.5 μm.
https://elifesciences.org/articles/73405/figures#video20

## Materials and methods

### Strains and culture conditions

The motile strains (WT) of *T. vulcanus* and its mutants were grown in BG-11 medium (*Stanier et al., 1971*) containing 20 mM Tris(hydroxymethyl)methyl-2aminoethanesulfonic acid(TES) (pH 7.5) as a buffer in moderate light (20 µmol m$^{-2}$ s$^{-1}$) at 45°C with shaking to an optical density of approximately 0.5–1.0 at 750 nm. The WT showing clear positive phototaxis on agar plates of BG11 was reisolated from the original strain NIES-2134 in the Microbial Culture Collection at the National Institute for Environmental Studies (NIES, https://mcc.nies.go.jp/) and used here as the standard strain. Likewise, a substrain of *T. vulcanus* showing negative phototaxis on agar plates of BG11 (WT_N) was spontaneously found and isolated from the original culture of the NIES-2134 strain.

### Strain construction

All primers, plasmids, and strains used in this study are listed in *Table 1*, *Table 2*, *Table 3* respectively. The plasmid for the disruption of the *sesC* gene was already reported in *Enomoto et al., 2015*. For the construction of the other gene disruption mutant strains, an antibiotic resistance cassette was introduced to cause a partial deletion within each ORF. The PCR fragments of a vector backbone, an antibiotic resistance cassette, and the upstream and downstream sequences of 2–3 kbp were assembled using assembly cloning (AQUA cloning) (*Beyer et al., 2015*). For introduction of the frameshift mutation in the *tll1859* gene, the upstream and downstream homologous sequences were amplified from the genomic DNA of WT_N. A kanamycin resistance cassette was inserted downstream of the *tll1859* gene without any deletion. For construction of the PilA1-Cys knockin strain (PilA1$^{S114C}$), a chloramphenicol resistance cassette was inserted downstream of the *pilA1* gene without any deletion. The overlapping primers containing the point mutation for S114C, were utilized for site-directed mutagenesis according to the protocol of the PrimeSTAR Max Basal Mutagenesis kit (TaKaRa). The sequence integrity was verified by Sanger sequencing. Transformations of *T. vulcanus* were performed according to *Iwai et al., 2004*. For preparation and cultivation of the mutant strains, antibiotics were added to the medium at the following concentrations: chloramphenicol, 5 µg ml$^{-1}$; kanamycin 80 µg ml$^{-1}$; and spectinomycin plus streptomycin, 10 plus 5 µg ml$^{-1}$. Complete segregation of the mutant alleles in the multiple copies of the chromosomal DNA was verified by colony PCR. For the *tll1859toN* and PilA1$^{S114C}$ mutants, the correct mutations were confirmed by Sanger sequencing of the PCR fragments amplified by colony PCR.

### Optical microscopy and data analyses

Cell behaviour on the glass surface was visualized under an inverted microscope (IX83; Olympus) equipped with ×10 objective lenses (UPLFLN10 × 2 PH, NA0.3; Olympus), a CMOS camera (Zyla 4.2; Andor, or DMK33U174; Imaging Source), and an optical table (ASD-1510T; JVI, Japan). The position of the cell was visualized by infrared light from a halogen lamp with a bandpass filter (FBH850/40; Thorlabs) at a fluence rate of 1 µmol m$^{-2}$ s$^{-1}$. The heat-absorbing filter was removed from the optical axis of the halogen lamp to obtain a higher wavelength of light. The signal from lateral light illumination was removed by a bandpass filter (FBH850/40; Thorlabs) in a filter block turret. The microscope stage was heated at 45°C with a thermoplate (TP-110R-100; Tokai Hit, Japan). Projections of the images were captured as greyscale images with the camera under 1-s resolution and converted into a sequential TIF file without any compression. All data were analysed by ImageJ 1.48v (rsb.info.nih.gov/ij/) and its plugins, TrackMate, particle tracker and multitracker.

For the bead assay, the cells and microbeads were visualized under an inverted microscope (IX83; Olympus) equipped with ×40 objective lenses (LUCPLFLN40 × PH, NA0.6; Olympus), a CMOS camera (Zyla 4.2; Andor, or DMK33U174; Imaging Source), and an optical table (ASD-1510T; JVI, Japan). The position of the cell and microbeads was visualized by a collimated blue-light LED (M450LP1; Thorlabs) through a dark-field condenser (U-DCD, NA0.8–0.92; Olympus) at a fluence rate of 200 µmol m$^{-2}$ s$^{-1}$. The microscope stage was heated at 45°C with a thermoplate (TP-110R-100; Tokai Hit). Projection of the image was captured as greyscale images with the camera under 0.1-s resolution and converted into a sequential TIF file without any compression.

For the direct visualization of fluorescent-labelled T4P filaments, the sample was examined under an inverted microscope (IX83; Olympus) equipped with a ×40 objective lens (LUCPLFLN40 × PH, NA0.6; Olympus), a filter set (GFP-4050B, Semrock), a CMOS camera (Zyla 4.2; Andor), and an optical

**Table 1.** Oligonucleotides used in this study.

| Name | Sequence (5′–3′) | Purpose |
|---|---|---|
| pUC19-5R_tvsesA | ATGATACGCGGGTGATTACCGAGCTCGAATTCAC | Δ*sesA* |
| pUC19-6F_tvsesA | GTGGCAAAACCACCTTTGCAAGCTTGGCGTAATC | |
| tvsesA-1F_pUC19 | GAATTCGAGCTCGGTAATCACCCGCGTATCATTG | |
| tvsesA-2R_Cm | TTCTATCAGCTGTCCCATCAATCCCCGAAACTGC | |
| Cm-3F_tvsesA | AGTTTCGGGGATTGATGGGACAGCTGATAGAAAC | |
| Cm-4R_tvsesA | ATGGGTGATATTGGCTTTTATCAGGCTCTGGGAG | |
| tvsesA-3F_Cm | CCCAGAGCCTGATAAAAGCCAATATCACCCATGC | |
| tvsesA-4R_pUC19 | TTACGCCAAGCTTGCAAAGGTGGTTTTGCCACAG | |
| pUC19-7R_tvsesB | TCAGGGGAGTCCAAAATACCGAGCTCGAATTCAC | Δ*sesB* |
| pUC19-8F_tvsesB | ACGGATTGCCATTGGTTGCAAGCTTGGCGTAATC | |
| tvsesB-1F_pUC19 | GAATTCGAGCTCGGTATTTTGGACTCCCCTGATG | |
| tvsesB-2R_Km | AGCAGGGGAATTGTTACACAGACTCTTCTGTGAC | |
| Km-5F_tvsesB | CACAGAAGAGTCTGTGTAACAATTCCCCTGCTCG | |
| Km-6R_tvsesB | GTTCAAGGTCCTCTTGCCAACCTTTCATAGAAGGC | |
| tvsesB-3F_Km | TTCTATGAAAGGTTGGCAAGAGGACCTTGAACTG | |
| tvsesB-4R_pUC19 | TTACGCCAAGCTTGCAACCAATGGCAATCCGTAG | |
| pUC19-21R_tvcikA | AGATGCGATTGTCCATTACCGAGCTCGAATTCAC | Δ*cikA* |
| pUC19-22F_tvcikA | TTCATTGGGCTTACAGTGCAAGCTTGGCGTAATC | |
| tvcikA-1F_pUC19 | GAATTCGAGCTCGGTAATGGACAATCGCATCTGC | |
| tvcikA-2R_Cm | TTCTATCAGCTGTCCCGCTGATAGCTACTGCTAG | |
| Cm-18F_tvcikA | AGCAGTAGCTATCAGCGGGACAGCTGATAGAAAC | |
| Cm-19R_tvcikA | AGATATTCCCGCTGTTTTTATCAGGCTCTGGGAG | |
| tvcikA-3F_Cm | CCCAGAGCCTGATAAAAACAGCGGGAATATCTGG | |
| tvcikA-4R_pUC19 | TTACGCCAAGCTTGCACTGTAAGCCCAATGAACC | |
| pUC19-19R_tvpixD | GTTACCATTCGCAAGTTACCGAGCTCGAATTCAC | Δ*pixD* |
| pUC19-20F_tvpixD | GCTAGTCCTGAACCAATGCAAGCTTGGCGTAATC | |
| tvpixD-1F_pUC19 | GAATTCGAGCTCGGTAACTTGCGAATGGTAACCG | |
| tvpixD-2R_Cm | TTCTATCAGCTGTCCCAATCAGGCGATGTAGTCC | |
| Cm-15F_tvpixD | ACTACATCGCCTGATTGGGACAGCTGATAGAAAC | |
| Cm-16R_tvpixD | CAGCATTCCCGTAATGTTTATCAGGCTCTGGGAG | |
| tvpixD-3F_Cm | CCCAGAGCCTGATAAACATTACGGGAATGCTGTG | |
| tvpixD-4R_pUC19 | TTACGCCAAGCTTGCATTGGTTCAGGACTAGCTG | |
| pUC19-23R_tvtll1282 | TTTGGCATCTTCAAGGTACCGAGCTCGAATTCAC | Δ*LOV* |
| pUC19-24F_tvtll1282 | TGAGCAAAGGGATCATTGCAAGCTTGGCGTAATC | |
| tvtll1282-1F_pUC19 | GAATTCGAGCTCGGTACCTTGAAGATGCCAAAGC | |
| tvtll1282-2R_Cm | TTCTATCAGCTGTCCCAGCATAGCGTTCCATCTC | |
| Cm-20F_tvtll1282 | GATGGAACGCTATGCTGGGACAGCTGATAGAAAC | |
| Cm-21R_tvtll1282 | AATAGGCTCAGGACATTTTATCAGGCTCTGGGAG | |
| tvtll1282-3F_Cm | CCCAGAGCCTGATAAAATGTCCTGAGCCTATTGG | |

*Table 1 continued on next page*

*Table 1 continued*

| Name | Sequence (5'–3') | Purpose |
|---|---|---|
| tvtll1282-4R_pUC19 | TTACGCCAAGCTTGCAATGATCCCTTTGCTCAGC | |
| pUC19-25R_tvHCP | TGAACTTTGCCCTTGTTACCGAGCTCGAATTCAC | Δ*OCP* |
| pUC19-26F_tvCTDH | GATAGGGAGATTGCATTGCAAGCTTGGCGTAATC | |
| tvHCP-1F_pUC19 | GAATTCGAGCTCGGTAACAAGGGCAAAGTTCAGC | |
| tvHCP-2R_Cm | TTCTATCAGCTGTCCCGCTCATCAAAACTGAGGG | |
| Cm-22F_tvHCP | CTCAGTTTTGATGAGCGGGACAGCTGATAGAAAC | |
| Cm-23R_tvCTDH | TCTTCAAAGGGTGGATTTTATCAGGCTCTGGGAG | |
| tvCTDH-1F_Cm | CCCAGAGCCTGATAAAATCCACCCTTTGAAGAGC | |
| tvCTDH-2R_pUC19 | TTACGCCAAGCTTGCAATGCAATCTCCCTATCCG | |
| pUC19-15R_tvpilB | CAGTTGAATCTGGGTTTACCGAGCTCGAATTCAC | Δ*pilB* |
| pUC19-16F_tvpilB | ACAGGTTTTGGAGGTCTGCAAGCTTGGCGTAATC | |
| tvpilB-1F_pUC19 | GAATTCGAGCTCGGTAAACCCAGATTCAACTGGG | |
| tvpilB-2R_Cm | TTCTATCAGCTGTCCCTTGTGGTACTGGCGTAAC | |
| Cm-11F_tvpilB | TACGCCAGTACCACAAGGGACAGCTGATAGAAAC | |
| Cm-12R_tvpilB | AGATCTTGAAGCGGGATTTATCAGGCTCTGGGAG | |
| tvpilB-3F_Cm | CCCAGAGCCTGATAAATCCCGCTTCAAGATCTTG | |
| tvpilB-4R_pUC19 | TTACGCCAAGCTTGCAGACCTCCAAAACCTGTAC | |
| pUC19-17R_tvpilT1 | TTTTTGCCAGCACCTTTACCGAGCTCGAATTCAC | Δ*pilT1* |
| pUC19-18F_tvpilT1 | TCTTGGCTGGCTTGATTGCAAGCTTGGCGTAATC | |
| tvpilT1-1F_pUC19 | GAATTCGAGCTCGGTAAAGGTGCTGGCAAAAAGC | |
| tvpilT1-2R_Cm | TTCTATCAGCTGTCCCAAGTCTGAACCACCGTTG | |
| Cm-13F_tvpilT1 | ACGGTGGTTCAGACTTGGGACAGCTGATAGAAAC | |
| Cm-14R_tvpilT1 | TGGCAAGCTGAATCGTTTTATCAGGCTCTGGGAG | |
| tvpilT1-3F_Cm | CCCAGAGCCTGATAAAACGATTCAGCTTGCCATC | |
| tvpilT1-4R_pUC19 | TTACGCCAAGCTTGCAATCAAGCCAGCCAAGAAG | |
| pUC19-11R_tvpilA1 | CTATTCTCTTTGCAGGTACCGAGCTCGAATTCAC | Δ*pilA1* |
| pUC19-12F_tvpilA1 | TGGTTTGGTGCCAGTATGCAAGCTTGGCGTAATC | |
| tvpilA1-1F_pUC19 | GAATTCGAGCTCGGTACCTGCAAAGAGAATAGCG | |
| tvpilA1-7R_Cm | TTCTATCAGCTGTCCCAGTTGCCGTCTGCTACAG | |
| Cm-32F_tvpilA1 | GTAGCAGACGGCAACTGGGACAGCTGATAGAAAC | |
| Cm-6R_tvpilA1 | GCAGTTGCACTATTGCTTTATCAGGCTCTGGGAG | |
| tvpilA1-3F_Cm | CCCAGAGCCTGATAAAGCAATAGTGCAACTGCTC | |
| tvpilA1-4R_pUC19 | TTACGCCAAGCTTGCATACTGGCACCAAACCATG | |
| pUC19-13R_tvHfq | CAGCGGGATGTGAATATACCGAGCTCGAATTCAC | Δ*hfq* |
| pUC19-14F_tvHfq | AAATGCGGCTTTCCTATGCAAGCTTGGCGTAATC | |
| tvHfq-1F_pUC19 | GAATTCGAGCTCGGTATATTCACATCCCGCTGTG | |
| tvHfq-2R_Cm | TTCTATCAGCTGTCCCTACTTGGCGAATGCTAGG | |
| Cm-9F_tvHfq | TAGCATTCGCCAAGTAGGGACAGCTGATAGAAAC | |
| Cm-10R_tvHfq | ACCTCGGTGCTATCTTTTTATCAGGCTCTGGGAG | |

*Table 1 continued on next page*

*Table 1 continued*

| Name | Sequence (5'–3') | Purpose |
|------|------------------|---------|
| tvHfq-3F_Cm | CCCAGAGCCTGATAAAAAGATAGCACCGAGGTTG | |
| tvHfq-4R_pUC19 | TTACGCCAAGCTTGCATAGGAAAGCCGCATTTGC | |
| pUC19-43R_tvtll1859 | ATCCATCGAACAAGCCTACCGAGCTCGAATTCAC | *tll1859toN* |
| pUC19-44F_tvtll1859 | GTCCCGTTCATTGATGTGCAAGCTTGGCGTAATC | |
| tvtll1859-1F_pUC19 | GAATTCGAGCTCGGTAGGCTTGTTCGATGGATTG | |
| tvtll1859-2R_Kmv2 | cctgagtgcttgcggcTTTCTAGGGGGTGTGAATC | |
| Km-21F_tvtll1859 | TCACACCCCCTAGAAAgccgcaagcactcagg | |
| Km-22R_tvtll1859 | TTGCAACACAGTTGTCcctttcatagaaggcggc | |
| tvtll1859-3F_Kmv2 | cgccttctatgaaaggGACAACTGTGTTGCAAACC | |
| tvtll1859-4R_pUC19 | TTACGCCAAGCTTGCACATCAATGAACGGGACTG | |
| pUC19-67F_tvpilA1 | gattggtttcaggcaaTGCAAGCTTGGCGTAATC | PilA1$^{S114C}$ |
| pUC19-68R_tvPilA1 | aacgatagggagacatTACCGAGCTCGAATTCAC | |
| tvPilA1-8F_pUC19 | GAATTCGAGCTCGGTAatgtctccctatcgttctag | |
| TvPilA1-9R_CmR | TTCTATCAGCTGTCCCttaagaacagttaggagcag | |
| Cm-44F_TvPilA1 | tcctaactgttcttaaGGGACAGCTGATAGAAAC | |
| Cm-45R_TvPilA1 | gttaattcctaaggtcTTTATCAGGCTCTGGGAG | |
| TvPilA1-10F_Cm | CCCAGAGCCTGATAAAgaccttaggaattaacagag | |
| TvPilA1-11R_pUC19 | TTACGCCAAGCTTGCAttgcctgaaaccaatcgg | |
| tvPilA1(S114C)–1F | caaacaTGCgctaatgatccccaaggg | |
| tvPilA1(S114C)–2R | attagcGCAtgtttgaacataagcatcattc | |

table (ASD-1510T; JVI, Japan). A blue laser beam (wavelength of 488 nm; OBIS488LS, Coherent) was introduced into the inverted microscope through a lens for epi-fluorescence microscopy. The blue laser beam was also introduced from the lateral side of the microscope stage through a prism for TIRF, and the cells were observed on the upper side of the chamber. The microscope stage was heated at 45°C with a thermoplate (TP-110R-100; Tokai Hit). Projection of the image was captured with the camera as a greyscale image with 0.1-s resolution and converted into a sequential TIF file without any compression.

## Phototaxis on glass with lateral light from LEDs

All procedures were performed at 45°C on a microscope stage heated with a thermoplate (TP-110R-100; Tokai Hit, Japan). The cell culture was poured into a tunnel chamber assembled by taping a coverslip (*Nakane and Nishizaka, 2017*), and both ends of the chamber were sealed with nail polish to keep from drying the sample. The position of the cell was visualized by infrared light from a halogen lamp with a bandpass filter (FBH850/40; Thorlabs) at a fluence rate of 1 µmol m$^{-2}$ s$^{-1}$. The cells were subjected to lateral light stimulus by an LED from the right side of the microscope stage at an angle of 5°. White LEDs at 20 and 500 µmol m$^{-2}$ s$^{-1}$ were used as moderate and strong light stimuli for phototaxis, respectively. Blue, teal, green, orange, red, and far-red light were applied by a monochromatic LED, M450LP1, M490L4, M530L3, M625L3, and M730L4 (Thorlabs), respectively. The LED light was collimated by the condenser lens and combined by dichroic mirrors (FF470-Di01, FF509-FDi01, FF560-FDi01, FF685-Di02; Semrock) to apply multicoloured light simultaneously. The wavelength of the resultant light was measured by a spectrometer (BIM-6002A, BroLight, China). Light intensity was measured with a power metre (Q82017A; Advantest, Japan).

**Table 2.** Plasmids used in this study.

| Name | Description | Reference/source |
|---|---|---|
| pUC19-ΔtvsesA_Cm | pUC19-based construct for knockout of *sesA* gene (NIES2134_109940) containing a chloramphenicol resistance cassette flanked by ~2500 bp regions upstream and downstream of *sesA*, resulting in ~2000 bp deletion within ORF; Cm[R], Amp[R] | This study |
| pUC19-ΔtvsesB_Km | pUC19-based construct for knockout of *sesB* gene (NIES2134_119260) containing a kanamycin resistance cassette flanked by ~2500 bp regions upstream and downstream of *sesB*, resulting in ~2000 bp deletion within ORF; Km[R], Amp[R] | This study |
| pS-Δtlr0911_Sp(F) | pCR-Script-based construct for knockout of *sesC* gene (NIES2134_110090) containing a spectinomycin/streptomycin resistance cassette flanked by ~1500 bp regions upstream and downstream of *sesC*, resulting in ~3400 bp deletion within ORF; Sp[R]/Sm[R], Amp[R] | *Enomoto et al., 2015* |
| pUC19-ΔtvcikA_Cm | pUC19-based construct for knockout of *cikA* gene (NIES2134_110210) containing a chloramphenicol resistance cassette flanked by ~2500 bp regions upstream and downstream of *cikA*, resulting in ~500 bp deletion within ORF; Cm[R], Amp[R] | This study |
| pUC19-ΔtvpixD_Cm | pUC19-based construct for knockout of *pixD* gene (NIES2134_124540) containing a chloramphenicol resistance cassette flanked by ~2500 bp regions upstream and downstream of *pixD*, resulting in ~100 bp deletion within ORF; Cm[R], Amp[R] | This study |
| pUC19-Δtvtll1282_Cm | pUC19-based construct for knockout of *LOV* gene (*tll1282*; NIES2134_112750) containing a chloramphenicol resistance cassette flanked by ~2500 bp regions upstream and downstream of *cikA*, resulting in ~2000 bp deletion within ORF; Cm[R], Amp[R] | This study |
| pUC19-ΔtvHCPCTDH_Cm | pUC19-based construct for knockout of *OCP* genes (HCP; NIES2134_112880 and CTDH; NIES2134_112890) containing a chloramphenicol resistance cassette flanked by ~2500 bp regions upstream and downstream of *OCP*, resulting in the deletion of the latter half of *HCP* ORF, the intergenic region, and the first half of *CTDH* ORF; Cm[R], Amp[R] | This study |
| pUC19-ΔtvpilB_Cm | pUC19-based construct for knockout of *pilB* gene (NIES2134_124120) containing a chloramphenicol resistance cassette flanked by ~2500 bp regions upstream and downstream of *pilB*, resulting in ~600 bp deletion within ORF; Cm[R], Amp[R] | This study |
| pUC19-ΔtvpilT1_Cm | pUC19-based construct for knockout of *pilT1* gene (NIES2134_124130) containing a chloramphenicol resistance cassette flanked by ~2500 bp regions upstream and downstream of *pilT1*, resulting in ~600 bp deletion within ORF; Cm[R], Amp[R] | This study |
| pUC19-ΔtvpilA1_Cm | pUC19-based construct for knockout of *pilA1* gene (NIES2134_108920) containing a chloramphenicol resistance cassette flanked by ~2500 bp regions upstream and downstream of *pilA1*, resulting in ~550 bp deletion within ORF and the promoter region; Cm[R], Amp[R] | This study |
| pUC19-ΔtvHfq_Cm | pUC19-based construct for knockout of *hfq* gene (NIES2134_124130) containing a chloramphenicol resistance cassette flanked by ~2500 bp regions upstream and downstream of *hfq*, resulting in ~10 bp deletion within ORF; Cm[R], Amp[R] | This study |
| pUC19-tvtll1859toN_Kmv2 | pUC19-based construct for introduction of the frameshift mutation present in WT_N within *tll1859* gene (NIES2134_117830) containing a kanamycin resistance cassette inserted after tll1859 ORF, flanked by ~2500 bp regions upstream and downstream of *tll1859* gene without any nucleotide deletion; Km[R], Amp[R] | This study |
| pUC19-tvpilA1(S114C)_Cm | pUC19-based construct for Cys knockin in *pilA1* gene (NIES2134_108920) containing a chloramphenicol resistance cassette inserted after *pilA1* ORF, flanked by ~2500 bp regions upstream and downstream of *pilA1* without any nucleotide deletion; Cm[R], Amp[R] | This study |

**Table 3.** Strains used in this study.

| Name | Description | Reference/source |
|---|---|---|
| *Thermosynechococcus vulcanus* WT | Isolated as wild type showing positive phototaxis towards moderate white light from NIES-2134 | Whole genome: AP018202 |
| *Thermosynechococcus vulcanus* WT_N | Isolated showing negative phototaxis towards moderate white light from NIES-2134 | Seq reads in DDBJ: DRA013349 |
| Δ*sesA* | *sesA* knockout mutant (Cm$^R$) | This study |
| Δ*sesB* | *sesB* knockout mutant (Km$^R$) | This study |
| Δ*sesC* | *sesC* knockout mutant (Sp$^R$/Sm$^R$) | *Enomoto et al., 2015* |
| Δ*cikA* | *cikA* knockout mutant (Cm$^R$) | This study |
| Δ*pixD* | *pixD* knockout mutant (Cm$^R$) | This study |
| Δ*LOV* | *LOV* knockout mutant (Cm$^R$) | This study |
| Δ*OCP* | *OCP* knockout mutant (Cm$^R$) | This study |
| Δ*pilB* | *pilB* knockout mutant (Cm$^R$) | This study |
| Δ*pilT1* | *pilT1* knockout mutant (Cm$^R$) | This study |
| Δ*pilA1* | *pilA1* knockout mutant (Cm$^R$) | This study |
| Δ*hfq* | *hfq* knockout mutant (Cm$^R$) | This study |
| *tll1859toN* | *tll1859* frameshift mutant (Km$^R$) | This study |
| PilA1$^{S114C}$ | *pilA1* Cys knockin mutant (Cm$^R$) | This study |

## Measurement of intracellular c-di-GMP concentration

Liquid cultures (1.5 ml each) of WT and Δ*sesA* strain of *T. vulcanus* were transferred to three prewarmed 2-ml tubes. Two tubes were used for the total nucleotide extraction, and the third tube was used for the determination of the protein concentration. The tubes were incubated under 70 µmol m$^{-2}$ s$^{-1}$ blue, green, red ($\lambda_{max}$=451, 528, and 625 nm), or white LED light (Philips LED tube 16W830) illumination at 45°C for 30 min. The cells were pelleted at 10,000 × *g* at 4°C for 30 s. For nucleotide extraction, cell pellets were immediately resuspended in 200 µl extraction solution (acetonitrile/methanol/water 2:2:1 [vol/vol/vol]), vortexed for 5 s, and heated at 95°C for 10 min. Samples were snap-cooled on ice and incubated for 15 min. The samples were centrifuged at 21,000 × *g* at 4°C for 10 min, and the supernatant was collected. The above extraction was repeated twice with 200 µl extraction solution each, without heat treatment. The combined supernatants (~600 µl) were incubated at −20°C overnight to precipitate proteins in the samples. The samples were centrifuged at 21,000 × *g* at 4°C for 10 min, and the supernatant was vacuum dried using SpeedVac at 42°C. Quantification of c-di-GMP was performed by HPLC/MS/MS analysis, as previously described (*Burhenne and Kaever, 2013*). For quantification of the total protein content, the pelleted cells were stored at −20°C. The de-frozen cells were suspended in 50 µl phosphate-buffered saline. A glass bead mix (0.1–0.11 and 0.25–0.5 mm) of ~0.7 volume was added. The tubes were vortexed vigorously for 60 s, snap-cooled at −80°C, and heated to 40°C for 10 min. The disruption was repeated once and incubated at RT until the beads were sedimented. Two microlitres of the supernatant was used for protein quantification using the Direct Detect system (Merck Millipore). The acquired concentrations of intracellular c-di-GMP were normalized to the total protein content.

## Genome resequencing of the phototactic negative *T. vulcanus* substrain

Genomic DNA of WT_N strain was prepared from a 40-ml culture with an optical density of ~2.0 with a Genomic-tip 20/G kit (Qiagen). Preparation of the paired-end DNA libraries and sequencing on the Illumina MiSeq platform were performed as described previously (*Hirose et al., 2021*). Raw sequence reads were deposited in the DDBJ Sequence Read Archive under accession number DRA013349. The low-quality region of the sequences was trimmed using fastq_quality_trimmer command with -t 20 and -l 36 options using FASTX-Toolkit (http://hannonlab.cshl.edu/fastx_toolkit/). Mapping of the

trimmed reads on the WT *T. vulcanus* genome (GenBank accession AP018202.2) was performed using bwa ver. 0.7.15 (*Li and Durbin, 2009*) and samtools ver. 1.3.1 (*Li et al., 2009*). Visualization of the mapped sequence reads was performed using Integrative Genomics Viewer ver. 2.8.10 (*Robinson et al., 2011*). Detection and annotation of the genomic variant were performed using Genome Analysis Toolkit ver. 3.8 (*Auwera GAVd, 2020*).

### Bead's assay for visualizing T4P dynamics

Fluorescent polystyrene beads 0.2 μm in size (FluoSpheres sulfate microspheres F8848, carboxylate-modified microspheres F8811, amine-modified microsphere F8764; Thermo Fisher) were diluted 300 times to 0.02% (wt/vol) in BG11 and used for the bead assay, as previously described (*Nakane and Nishizaka, 2017*). A coverslip was coated with 0.2% (vol/vol) collodion in isoamyl acetate and air-dried before use. The cell culture was poured into a tunnel chamber assembled by taping a coverslip. After incubation at 45°C for 2 min on the microscope stage, the cells were subjected to vertical illumination from blue-light LED through a dark-field condenser at a fluence rate of 200 μmol m$^{-2}$ s$^{-1}$. After illumination for 2 min, fluorescent beads were added to the sample chamber, and their movement was visualized by blue light illumination at 0.1-s intervals. Lateral illumination from green-light LEDs was applied for 2 min before adding the beads if needed.

### PilA labelling for fluorescence microscopy

Pilus filaments of the PilA1$^{S114C}$ mutant were labelled using maleimide-conjugated fluorophores (DyLight488 maleimide 46602, Thermo Fisher). Cell culture with a volume of 0.5 ml in BG11 medium at an OD750 of ~0.5 was mixed with 0.5 ml of 50 mM potassium phosphate buffer (pH 7.0), to which 1 μl Dylight488 maleimide dye (10 μg/μl in DMSO) was added. The cell suspensions were incubated at 45°C for 30 min, and then washed three times in BG11. The cell suspensions in BG11 were poured into a tunnel chamber assembled by taping a coverslip, and observed under fluorescence microscopy.

### Electron microscopy

Samples bound to the grids were stained with 2% (wt/vol) ammonium molybdate and observed by transmission electron microscopy, as previously described (*Nakane and Nishizaka, 2017*). Carbon-coated EM grids were prepared by a vacuum evaporator (VE-2012; Vacuum Device, Japan). Cells were placed on an EM grid and kept at 45°C in moderate light (20 μmol m$^{-2}$ s$^{-1}$) for 3 min. The cells were chemically fixed with 1% (vol/vol) glutaraldehyde in BG-11 for 10 min at RT. After washing three times with BG-11, the cells were stained with 2% ammonium molybdate and air-dried. Samples were observed under a transmission electron microscope (JEM-1400, JEOL) at 100 kV. The EM images were captured by a charge-coupled device camera.

## Acknowledgements

The authors thank Dr. Masahiko Ikeuchi (The University of Tokyo) for supplying the *T. vulcanus* wild-type strains. This study was supported in part by KAKENHI (16H06230, 20H05543, 21K07020) to DN, by funds from the Nakajima Foundation, the Noguchi Institute to DN, by KAKENHI (17K15244) to GE, by the German Science Foundation (WI2014/7-1, WI2014/8-1 and in frame of the SFB1381 – 403222702-SFB 1381 (A2)) to AW. GE is supported by the EMBO postdoctoral fellowship (ALTF 274-2017) and JSPS Overseas Research Fellowships.

## Additional information

### Funding

| Funder | Grant reference number | Author |
| --- | --- | --- |
| Japan Society for the Promotion of Science | 20H05543 | Daisuke Nakane |
| Japan Society for the Promotion of Science | 21K07020 | Daisuke Nakane |

| Funder | Grant reference number | Author |
|---|---|---|
| Japan Society for the Promotion of Science | 16H06230 | Daisuke Nakane |
| Nakajima Foundation | | Daisuke Nakane |
| Noguchi Institute | | Daisuke Nakane |
| Japan Society for the Promotion of Science | 17K15244 | Gen Enomoto |
| Deutsche Forschungsgemeinschaft | WI2014/7-1 | Annegret Wilde |
| Deutsche Forschungsgemeinschaft | WI2014/8-1 | Annegret Wilde |
| EMBO postdoctoral fellowship | ALTF 274-2017 | Gen Enomoto |
| Japan Society for the Promotion of Science | Overseas Research Fellowships | Gen Enomoto |
| Deutsche Forschungsgemeinschaft | 403222702-SFB 1381 (A2) | Annegret Wilde |

The funders had no role in study design, data collection, and interpretation, or the decision to submit the work for publication.

## Author contributions
Daisuke Nakane, Conceptualization, Formal analysis, Funding acquisition, Investigation, Methodology, Writing – original draft, Writing – review and editing; Gen Enomoto, Conceptualization, Funding acquisition, Investigation, Methodology, Resources, Writing – original draft, Writing – review and editing; Heike Bähre, Yuu Hirose, Investigation, Methodology; Annegret Wilde, Conceptualization, Funding acquisition, Writing – original draft, Writing – review and editing; Takayuki Nishizaka, Conceptualization

## Author ORCIDs
Daisuke Nakane http://orcid.org/0000-0002-8201-2608
Gen Enomoto http://orcid.org/0000-0002-9492-7557
Annegret Wilde http://orcid.org/0000-0003-0935-8415

## Decision letter and Author response
Decision letter https://doi.org/10.7554/eLife.73405.sa1
Author response https://doi.org/10.7554/eLife.73405.sa2

## Additional files

### Supplementary files
• Transparent reporting form

### Data availability
All data generated or analysed during this study are included in the manuscript and supporting file.

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
