## [Editor Report]

In this fascinating study, the authors explore the cellular and molecular bases of phototaxis in Cynaobacteria. It is shown that phototaxis is a highly efficient Spatio-temporal response involving "within a pole regulation" of retractile Type-IV pili. This unique regulation involves the interplay of three possible photoreceptors that regulate the intracellular concentration of c-di-GMP in response to green to blue light transitions.

---

## [Decision Letter]

**Decision letter after peer review:**

Thank you for submitting your article "*Thermosynechococcus* switches the direction of phototaxis by a c-di-GMP dependent process with high spatial resolution" for consideration by *eLife*. Your article has been reviewed by 2 peer reviewers, and the evaluation has been overseen by Tâm Mignot as a Reviewing Editor and Gisela Storz as the Senior Editor. The following individual involved in review of your submission has agreed to reveal their identity: Yuki Inclan (Reviewer #2).

Essential revisions:

1. While both reviewers agree that the work is potentially of broad significance, they insist that mechanistic links are currently missing to make the claim that the identified SesABC connect to Type-IV pili (TFP) activity via the increase intracellular cdiGMP. While it is understood that identifying a cdiGMP receptor for TFP function is probably beyond the scope of the present manuscript, a significant additional body of evidence linking cdiGMP levels to TFP activity in the context of phototaxis is deemed essential for acceptance of this manuscript.

2. The observation that the cells are able to steadily move along the light axis but perpendicular to their long axis is very interesting considering the T4P appear to be bipolarly localized. However, there is currently little proposed mechanism. It is possible that subpolar pili are activated but it also possible that consecutive activation of the pili at opposite poles lead to biased motions. Showing single cell videos and a more thorough analysis of T4P filament dynamics depending on positive/negative phototaxis and according to the angle of the cell would provide necessary additional insights.

*Reviewer #2 (Recommendations for the authors):*

The authors use a custom optical set up to visualize individual cells and microcolonies of T. vulcanus phototaxing towards teal/green light and away from blue-green light. This directed motility is dependent on T4P. They went on to identify the photoreceptors, SesA, SesB, and SesC which are required for negative phototaxis and these photoreceptors contain c-di-GMP synthesis and degradation domains. They showed that the concentration of the second messenger c-di-GMP correlates with the direction of phototaxis, with higher levels of c-di-GMP resulting in negative phototaxis. They also observed that the rod-shaped cells moving along the incident light axis can move consistently perpendicular to their long axis during phototaxis and suggest that this observation shows a polarity regulation coined "within-a-pole" polarity regulation of the polarly localized T4P.

The conclusions of this paper are mostly supported by the data. A major strength of this study is the careful microscopy and analysis. The experiments are clearly described in the text and the results are communicated well in the figures. The kymographs and videos are striking with the results of cells phototaxing towards green light and away from green light when blue light is present. The assays with the changing wavelengths and cell velocity responses are very clear. It is convincing that T4P are required for phototaxis of T. vulcanus.

The genetic link to phototaxis showing that SesA, SesB and SesC is convincing, but some details could strengthen the link. For example, the microcolony aggregation is an interesting but confounding factor. The conclusions would be strengthened if the authors could decouple the microcolony formation from phototaxis.

The observation that the cells are able to steadily move along the light axis but perpendicular to their long axis is very interesting considering the T4P appear to be bipolarly localized. There is some discussion on the micro-optic effect in single cells but it does not include the observation that the negative phototaxis to green light occurs no matter where the direction of blue light comes from or the micro-optic effect in a microcolony.

This work nicely illustrates how observing single cells can inform macroscopic phenotypes in the lab or in natural habitats.

This is wonderful work and I only have a few comments and questions about the work and potential experiments:

It appears that the higher the fluence of blue light, the higher the velocity of the cells and the more microcolonies are formed. Is there a dependence on the microcolony size to the velocity, or the pili numbers/density? Do the cells move faster with more pili and do those cells also have more c-di-GMP? Is anything known about the transcriptional regulation of the T4P in T. vulcanus?

In that same vein, for Figure 6-supplement 2, Is there a dependence on the velocity of the pili extension/retraction with increasing fluence? Or, why does the velocity of the cells increase with increasing fluence as shown in Figure 2 —figure supplement 5.

The link between SesA, SesB and SesC and negative phototaxis could be strengthened if the authors can decouple microcolony formation and negative phototaxis. The reference Enomoto, PNAS 2015 describes a PilZ protein T110007 involved in aggregation. Can the authors use a mutant to abolish the cell aggregation phenotype and only look at single cells? In addition, if a DGC is expressed from a heterologous expression system in a ∆sesA mutant, will that non-specific increase in c-di-GMP rescue the negative phototaxis phenotype? And conversely if a PDE is expressed in wt, will negative phototaxis still occur in the presence of blue light?

*Reviewer #3 (Recommendations for the authors):*

In this manuscript Nakane et al., investigate the phototaxis of the rod shaped bacteria Thermosynechococcus vulcanus. This is important because most our knowledge on phototaxis bacteria originates from study on round shaped cell. More precisely, the author's firstly convincingly demonstrate a positive and a negative phototaxis in T. vulcanus, regulated by a green to blue light ratio. Secondly, the author's show the implication of SesABC proteins in the negative phototaxis regulation, via a potential modulation of the C-di-GMP level in the cell. Finally, the author's characterised the T4P-dependant motility as the output for phototactic behaviours in T. vulcanus. The authors further investigated the T4P dynamics during negative phototaxis using fluorescent beads.

Overall the work present in the manuscript is well carry out and conclusions drawn from experiments are largely correct. Nevertheless, my general impression is that the proposed manuscript is at this stage premature. The present work needs further experiments to strength the actual conclusions but also to get more molecular details. Ultimately, a final working model would be beneficial to summarise the proposed molecular links from SesABC to T4P regulation.

I – The author's attribute the defect of negative phototaxis observed in the SesA mutant to the level of C-di-GMP in the cell, mainly because a SesA mutant shows a two fold decrease in C-di-GMP concentration upon blue light treatment. However, this measurement has been realised in a batch culture and normalised to dry cell mass. At the opposite, the negative phototaxis observed at single cell level occurs in a range of less than a minute (Figure 2). It would be therefore important for the authors to strength the implication of C-di-GMP in the phototaxis regulation. For example, the authors could ectopically modulate the level of C-di-GMP in the cell, via the expression of ectopic a diguanylate cyclase or phosphodiesterase enzymes, and observe its effect on phototaxis.

II – The author's used fluorescent beads to visualize T4P dynamics. As it was previously described, the author's show that it is specific of the T4P activity and it also can reveal T4P retraction. Then, the author's used this method to convincingly show that cells that move perpendicular of the light source have only active pili at one half of the both cell poles (Fig6). It is an interesting observation but again it gets short of details.

– The manuscript would definitively benefit from more general analysis of T4P dynamics during phototaxis. For example, during the switch from positive to negative phototaxis. What are the behaviours (T4P pole activation) of cells parallel to the light source?

– Besides, as suggested by the authors in the discussion, having the intracellular localisation of the Atpase PilB would definitively be a plus.

– Moreover, in the Discussion section the author proposed the existence of "a specific signalling system with high special resolution" to explain the asymmetric polar T4P activation. Why could it not be a molecular mechanism similar to the one observed in round cell such as Synechocystis, where the light receptor PixD regulates T4P function at some part of the cell according to the direction of the light.

III – The links between the C-di-GMP concentration and T4P dynamics during the switch from positive to negative phototaxis is absent. The author's proposed in the discussion a potential binding of C-di-GMP to PilB as previously shown for some T4P. Could it be tested here by the author's since they seem to be able to handle C-di-GMP?

1 – What is the motility phenotype on plate of the different ses mutants as in Figure 1A?

2 – It would be interesting for the reader to have a sup video a pil mutant during phototaxis.

3 – Line 228. The authors conclude that T. vulacanus has a bipolar localisation of T4P. At this stage in the manuscript, I don't see how the authors can conclude this, since only one pole is presented.

4 – Line 255. The data showing that in a pilB mutant beads do not accumulate is not referenced in the text.

5 – As a general idea, since T. vulcanus is genetically tractable, could it be feasible to realise a Tn-seq combined with a batch motility selection upon light similar to Figure 1A to uncover other phototaxis regulators?

---

## [Author Response]

Essential revisions:1. While both reviewers agree that the work is potentially of broad significance, they insist that mechanistic links are currently missing to make the claim that the identified SesABC connect to Type-IV pili (TFP) activity via the increase intracellular cdiGMP. While it is understood that identifying a cdiGMP receptor for TFP function is probably beyond the scope of the present manuscript, a significant additional body of evidence linking cdiGMP levels to TFP activity in the context of phototaxis is deemed essential for acceptance of this manuscript.

We have performed additional experiments, which are summarized in Figure 4. We were able to isolate a spontaneous mutant that showed constitutive negative phototaxis under white light. Genome sequencing identified a frameshift mutation in the *tll1859* gene, which encodes a GGDEF/EAL domain protein (Figure 4AB). Introducing the same mutation in our original wild type confirmed that this frameshift mutation is responsible for negative phototaxis even under green light irradiation (Figure 4C-G). Further, we have measured a higher c-di-GMP concentration in both mutant strains in comparison to the wild type, suggesting that Tll1859 has phosphodiesterase activity. This additional experimental evidence independently supports our idea that increased c-di-GMP levels switch the direction of T4P-dependent phototaxis from positive to negative in *Thermosynechococcus*. We added a paragraph in the Results part to explain these experiments.

2. The observation that the cells are able to steadily move along the light axis but perpendicular to their long axis is very interesting considering the T4P appear to be bipolarly localized. However, there is currently little proposed mechanism. It is possible that subpolar pili are activated but it also possible that consecutive activation of the pili at opposite poles lead to biased motions. Showing single cell videos and a more thorough analysis of T4P filament dynamics depending on positive/negative phototaxis and according to the angle of the cell would provide necessary additional insights.

We have performed additional experiments, which are summarized in Figure 8 and Videos 17-20. Most importantly, we succeeded in direct visualization of T4P filaments in living cells of a PilA1–cysteine knock-in mutant using a maleimide-conjugated dye. This method was recently established for T4P of several heterotrophic bacteria. Our new experiments clearly show that the *Thermosynechococcus* T4P filaments are bipolarly localized and assemble and retract at both cell poles simultaneously (Figure 8). When the cells move perpendicular to their long axis, the T4P filaments at both poles show a biased distribution in the direction of cellular movement (Figure 8EF). These results support our idea that the T4P are indeed asymmetrically activated within a single cell pole. We have added a chapter in the Results part for these experiments and also modified the text to explain our ideas more clearly.

Reviewer #2 (Recommendations for the authors):[…]The observation that the cells are able to steadily move along the light axis but perpendicular to their long axis is very interesting considering the T4P appear to be bipolarly localized. There is some discussion on the micro-optic effect in single cells but it does not include the observation that the negative phototaxis to green light occurs no matter where the direction of blue light comes from or the micro-optic effect in a microcolony.

We have added the following sentences in the Discussions:

“The focused green light would excite yet unknown photosensory molecules to induce spatially localized signalling, whereas the position of the focused blue light is not crucial for directional switching. As we showed, the direction of blue light illumination did not influence directionality of movement, because cells do not move in random orientation (Figure 2 —figure supplement 6). Thus, blue light does not control the directional light-sensing capability, instead it provides the signal for the switch between positive and negative phototaxis. This is very similar to the situation in Synechocystis where the blue light receptor PixD controls the switch between negative and positive phototaxis independently of the position of the blue-light source (Sugimoto et al., 2017).”

This work nicely illustrates how observing single cells can inform macroscopic phenotypes in the lab or in natural habitats.This is wonderful work and I only have a few comments and questions about the work and potential experiments:It appears that the higher the fluence of blue light, the higher the velocity of the cells and the more microcolonies are formed. Is there a dependence on the microcolony size to the velocity, or the pili numbers/density? Do the cells move faster with more pili and do those cells also have more c-di-GMP? Is anything known about the transcriptional regulation of the T4P in T. vulcanus?

We appreciate your interesting comments. At the moment, it would be too early to conclude about direct correlations between light intensity, velocity, pili numbers, and microcolony size. The cells show directional switching from positive to negative phototaxis when the ratio of blue light to green light was increased (Figure 2 —figure supplement 5E). The effect of a higher blue-to-green light ratio is not consistent. The negative phototaxis response was fast and directional at a ratio of 200/70 μmol m^−2^ s^−1^ blue/green light and this ratio also led to the formation of microcolonies. However, the directionality of phototaxis was decreased when the blue-light portion was increased by a factor of 10 (Figure 2 —figure supplement 5E). In contrast, microcolony formation was enhanced under these conditions (Figure 2 —figure supplement 5A). We have added the following sentence:

“At more than 700 μmol m^−2^ s^−1^ blue light, however, the directionality of negative phototaxis became decreased (Figure 2 —figure supplement 5E). In contrast, microcolony formation was enhanced under these conditions (Figure 2 —figure supplement 5A).”

In general, we think that we have created some confusion by the term “velocity,” which we have used to describe the distance which cells traveled between point A and point B along the light axis. This “velocity” depends on the speed of the cells as well as on how direct the cells move from A to B. Therefore, we have changed “velocity” to the term “displacement” which we think describes now more clearly this characteristic. In addition, there seems to be no direct correlation between displacement and c-di-GMP concentration. In our new experiment using mutants with an overall higher c-di-GMP content (Figure 4), there was no effect on the mean displacement of cells. Therefore, we think that c-di-GMP seems to be a signal for the directional switch, but not for velocity. Whether cells would move faster with more pili, we cannot say at the moment. Most probably, there is a trade-off between velocity, directionality, number (but also length) of pili, and microcolony formation. In addition, the production of more pili requires more motor ATPases and more coordination to ensure efficient extension and retraction dynamics of the pili. The pilus labeling methods we have established in the revised version of this manuscript might be the key to answer all these questions in the future.

In other cyanobacteria, various signaling inputs are transduced to the transcriptional regulation of type IV pili components, including circadian rhythm (Taton et al., Nat Commun, 2020) and cAMP (Yoshimura et al., Plant Cell Physiol, 2002). Notably, c-di-GMP also controls the transcription of several minor pilins in Synechocystis sp. PCC 6803 (Wallner et al., Photochem. Photobiol. Sci., 2020). These previous reports suggested that minor components of the pili filaments may govern cellular behavior, such as motility and biofilm formation. However, we have no data on transcriptional regulation of T4P genes in Thermosynechococcus and the role of minor pilins in this organism. Therefore, in the future, we plan to reveal the molecular events of c-di-GMP-induced directional switching, including the composition of the Thermosynechococcus type IV pili filaments, in more detail.

In that same vein, for Figure 6-supplement 2, Is there a dependence on the velocity of the pili extension/retraction with increasing fluence? Or, why does the velocity of the cells increase with increasing fluence as shown in Figure 2 —figure supplement 5.

Actually, we do not show that velocity (displacement) increases with blue light intensity. In Figure 2 – supplement 5E, cells switch between positive and negative phototaxis, but the mean displacement is more or less the same (+30 versus −30). It even decreases from −30 to around −20 when the intensity of the blue light increases (from 200 to 2,000 μmol m^−2^ s^−1^)*.* It would be very challenging for us to assess the dependency of the velocity of the pili dynamics on the light intensity. We are not able to detect the movement of the beads without the application of strong light illumination in our experimental setup. However, to address this question, we have performed additional experiments to measure the displacement of single cells, which are summarized in the new Figure 8 and Videos 17-20. The cells show the asymmetric distribution of T4P filaments within a cell pole during directional movement. This increased asymmetry of pilus activation at both poles might cause an overall increase of cell displacement during phototaxis. We have added a chapter in the Results part for these experiments.

The link between SesA, SesB and SesC and negative phototaxis could be strengthened if the authors can decouple microcolony formation and negative phototaxis. The reference Enomoto, PNAS 2015 describes a PilZ protein T110007 involved in aggregation. Can the authors use a mutant to abolish the cell aggregation phenotype and only look at single cells? In addition, if a DGC is expressed from a heterologous expression system in a ∆sesA mutant, will that non-specific increase in c-di-GMP rescue the negative phototaxis phenotype? And conversely if a PDE is expressed in wt, will negative phototaxis still occur in the presence of blue light?

Thank you for these interesting ideas. However, we were not able to decouple microcolony formation and negative phototaxis so far. Note that the short-term microcolony formation under the microscope and the long-term cell aggregation seem to be two independent phenomena. We previously showed that the ∆*sesA* mutant cannot form macroscopic cell aggregation (Enomoto et al., PNAS, 2015), but it showed microcolony formation under microscopy even in the absence of negative phototaxis. This result suggests that blue light-induced microcolony formation seems to be independent of the c-di-GMP signaling network. It is a future challenge to address what molecular events govern the microcolony/biofilm formation of this cyanobacterium.

Utilizing heterologous expressions in *T. vulcanus* is challenging for yet unknown reasons. In addition, the functionality of heterologously expressed proteins is sometimes impaired, possibly due to the high-temperature cultivation at 45°C. However, we have got another independent experimental evidence for the role of c-di-GMP as a regulator of phototaxis direction. By comparative genomics, we identified a frameshift mutation in a putative phosphodiesterase gene that confers an increase of the intracellular c-di-GMP concentration (see our answer to the editor’s comment 1). These mutants showed negative phototaxis under conditions where WT cells showed positive phototaxis (Figure 4). We have added a paragraph in the Results part for these experiments.

Reviewer #3 (Recommendations for the authors):In this manuscript Nakane et al., investigate the phototaxis of the rod shaped bacteria Thermosynechococcus vulcanus. This is important because most our knowledge on phototaxis bacteria originates from study on round shaped cell. More precisely, the authors firstly convincingly demonstrate a positive and a negative phototaxis in T. vulcanus, regulated by a green to blue light ratio. Secondly, the author's show the implication of SesABC proteins in the negative phototaxis regulation, via a potential modulation of the C-di-GMP level in the cell. Finally, the author's characterised the T4P-dependant motility as the output for phototactic behaviours in T. vulcanus. The authors further investigated the T4P dynamics during negative phototaxis using fluorescent beads.Overall the work present in the manuscript is well carry out and conclusions drawn from experiments are largely correct. Nevertheless, my general impression is that the proposed manuscript is at this stage premature. The present work needs further experiments to strength the actual conclusions but also to get more molecular details. Ultimately, a final working model would be beneficial to summarise the proposed molecular links from SesABC to T4P regulation.I – The author's attribute the defect of negative phototaxis observed in the SesA mutant to the level of C-di-GMP in the cell, mainly because a SesA mutant shows a two fold decrease in C-di-GMP concentration upon blue light treatment. However, this measurement has been realised in a batch culture and normalised to dry cell mass. At the opposite, the negative phototaxis observed at single cell level occurs in a range of less than a minute (Figure 2). It would be therefore important for the authors to strength the implication of C-di-GMP in the phototaxis regulation. For example, the authors could ectopically modulate the level of C-di-GMP in the cell, via the expression of ectopic a diguanylate cyclase or phosphodiesterase enzymes, and observe its effect on phototaxis.

We highly appreciate your evaluation and comments. As we pointed out in our response to reviewer 2, utilizing heterologous expression systems in *T. vulcanus* is challenging, maybe due to the cultivation of cells at of 45°C. However, we were lucky in isolating a spontaneous mutant (named WT_N) that shows constitutive negative phototaxis under lateral light illumination. By comparative genomics, we identified the frameshift mutation that confers an increase of the intracellular concentration of c-di-GMP and which was accompanied by negative phototaxis under the condition where the WT cells showed positive phototaxis (Figure 4). We have added a paragraph in the Results part for these experiments. See also our comments to the other reviewer and the editor concerning these new experiments, which support the role of c-di-GMP in directional switching. In addition, the figure formerly assigned as Figure 3 —figure supplement 1 was moved to the main manuscript as Figure 3C, because we think that the data of the intracellular concentration of c-di-GMP are very important to support our conclusions.

II – The author's used fluorescent beads to visualize T4P dynamics. As it was previously described, the author's show that it is specific of the T4P activity and it also can reveal T4P retraction. Then, the author's used this method to convincingly show that cells that move perpendicular of the light source have only active pili at one half of the both cell poles (Fig6). It is an interesting observation but again it gets short of details.– The manuscript would definitively benefit from more general analysis of T4P dynamics during phototaxis. For example, during the switch from positive to negative phototaxis. What are the behaviours (T4P pole activation) of cells parallel to the light source?– Beside, as suggested by the author's in the discussion, having the intracellular localisation of the Atpase PilB would definitively be a plus.– Moreover, in the Discussion section the author proposed the existence of "a specific signalling system with high special resolution" to explain the asymmetric polar T4P activation. Why could it not be a molecular mechanism similar to the one observed in round cell such as Synechocystis, where the light receptor PixD regulates T4P function at some part of the cell according to the direction of the light.

In order to get more direct insights into T4P dynamics, we have performed additional experiments, which are summarized in Figure 8 and Videos 17-20. Importantly, we succeeded in visualizing T4P filaments by PilA1 labelling using live cells. The T4P filaments were bipolarly localized and showed dynamics of assembly and retraction at both cell poles. When the cells moved perpendicular to their long axis, the T4P filaments at both poles showed biased distribution towards the same direction of cellular movement. These results support our idea that T4P are asymmetrically activated within a single cell pole. This asymmetric activation can rely on the localization of PilB ATPase. We would like to address how a molecular machinery such as PilB governs directional switching events. However, GFP-tagging has not been established in thermophilic cyanobacteria so far. We have added a chapter in the Results part for these experiments. Please, also pay attention to our answers to similar comments of the other reviewer.

Our results suggest that the *T. vulcanus* cell can actuate the spatially resolved signaling even within a cell pole to activate the pilus activity at only one side of a cell pole to enable biased cellular movements. This finding means that the cell harnesses "a specific signalling system with high special resolution" compared to other rod-shaped bacteria showing pole-to-pole regulation of cell polarity. We do not exclude that a system which works similar to the PixD/PixE complex in *Synechocystis* contributes to the asymmetric localization of the pili in *Thermosynechococcus* motility. *Thermosynechococcus* encodes a PixD protein but no PixE homolog. For *Synechocystis*, it was shown very recently that PATAN domain response regulators (including PixE) bind PilB1 and PilC and can switch the direction of movement (Han et al., Mol. Microbiol. 2021). *Thermosynechococcus* encodes homologs of such PATAN-domain response regulators, but at the moment, we do not know whether they have a similar function in both cyanobacteria.

III – The links between the C-di-GMP concentration and T4P dynamics during the switch from positive to negative phototaxis is absent. The author's proposed in the discussion a potential binding of C-di-GMP to PilB as previously shown for some T4P. Could it be tested here by the author's since they seem to be able to handle C-di-GMP?

The experimental verification of the binding of c-di-GMP to PilB is ongoing work, but it seems that direct binding of c-di-GMP to PilB is either very weak or does not happen in our setup. Thus, detailed molecular events of c-di-GMP signaling are out of the scope of the current study. However, we do show in the revised version of the manuscript that pilus extension and retraction dynamics are not different between positive and negative phototaxis (Figure 7 − figure supplement 2), suggesting that c-di-GMP most probably does not affect the activity of the PilB protein. Therefore, we have modified the sentence about the binding of c-di-GMP to PilB in the Discussion part as follows:

“Since we did not observe a change in pilus dynamics under green and green/blue light illumination (Figure 7 − figure supplement 2), the T4P regulation in T. vulcanus may not be explained simply by a specific activation of PilB (Floyd et al., 2020, Hendrick et al., 2017).”

In addition, we have performed experiments to show additional data that the c-di-GMP levels switch the direction of T4P-dependent phototaxis (new Figure 4). We also performed additional experiments to visualize T4P dynamics by PilA labeling (new Figure 8), which suggest asymmetric activation of pili and most probably of the motor ATPases as well.

1 – What is the motility phenotype on plate of the different ses mutants as in Figure 1A?

The ses mutants showed positive phototaxis on agar plates under lateral green light illumination. Generally, phototaxis assays on agar plates take at least several hours for observation. Since long time-scale responses are rather complicated to control, because of various changing parameters, such as media compositions and self-shading effects, we focused on the cell behavior in a short time scale in this work. Therefore, data on long time-scale responses are not discussed in this manuscript.

2 – It would be interesting for the reader to have a sup video a pil mutant during phototaxis.

We have added the video of Δ*pilB* mutant cells under green light illumination (Video 11).

3 – Line 228. The authors conclude that T. vulacanus has a bipolar localisation of T4P. At this stage in the manuscript, I don't see how the authors can conclude this, since only one pole is presented.

We have modified the phrases as follows. “T4P filaments were polarly localized in T. vulcanus.” In addition, the bipolar localization has been confirmed by direct visualization of T4P by PilA labeling in the new Figure 8.

4 – Line 255. The data showing that in a pilB mutant beads do not accumulate is not referenced in the text.

We have added the phrases to make this point clearer: “This accumulation was not observed in the ΔpilB mutant (Figure 6 —figure supplement 1A).”

5 – As a general idea, since T. vulcanus is genetically tractable, could it be feasible to realise a Tn-seq combined with a batch motility selection upon light similar to Figure 1A to uncover other phototaxis regulators?

Thank you very much for the insightful suggestion. We think that this would be a possible approach, though transposon mutagenesis has never been applied to *Thermosynechococcus,* and it is possible that we will not be able to select single non-moving colonies because non-movers can be trapped in moving microcolonies.